# MetaLA: Unified Optimal Linear Approximation to Softmax Attention Map

**Yuhong Chou**[1,2]*, **Man Yao**[2]*, **Kexin Wang**[2], **Yuqi Pan**[2], **Ruijie Zhu**[3],
**Yiran Zhong**[4], **Yu Qiao**[4], **Jibin Wu**[1], **Bo Xu**[2], **Guoqi Li**[2]†

[1]The Hong Kong Polytechnic University
[2]Institute of Automation, Chinese Academy of Sciences
[3]UC Santa Cruz
[4]Shanghai AI Lab

## Abstract

Various linear complexity models, such as Linear Transformer (LinFormer), State Space Model (SSM), and Linear RNN (LinRNN), have been proposed to replace the conventional softmax attention in Transformer structures. However, the optimal design of these linear models is still an open question. In this work, we attempt to answer this question by finding the best linear approximation to softmax attention from a theoretical perspective. We start by unifying existing linear complexity models as the linear attention form and then identify three conditions for the optimal linear attention design: i) Dynamic memory ability; ii) Static approximation ability; iii) Least parameter approximation. We find that none of the current linear models meet all three conditions, resulting in suboptimal performance. Instead, we propose Meta Linear Attention (MetaLA) as a solution that satisfies these conditions. Our experiments on Multi-Query Associative Recall (MQAR) task, language modeling, image classification, and Long-Range Arena (LRA) benchmark demonstrate that MetaLA is more effective than the existing linear models. Code: https://github.com/BICLab/MetaLA

## 1 Introduction

Transformer with softmax attention [1] benefits from efficient parallel training and exhibits impressive performance on deep learning applications [2, 3, 4, 5, 6, 7], but it suffers from the quadratic growth of computation cost to the input length [8]. Linear recurrent models, such as LinFormer [9], SSM [10], and LinRNN [11], are expected to achieve linear substitution of Transformer. The original intention of LinFormer is to replace softmax attention, which exploits the kernel approach to decompose softmax operation; typical work includes TransNormer [12, 13], RetNet [14], GLA [15]. On the other hand, SSMs, such as S4 [10] and Mamba [16], are models inspired by the classical state-space approach, which enjoys sub-quadratic training and inference like either a recurrence or convolution. In contrast, LinRNN is a revival of traditional RNNs, including RWKV-4 [17], Griffin [18], LRU [19], etc., which solves the training difficulties of traditional RNNs due to nonlinear dependencies between hidden states. It is natural to think that they are different types of models, since these LinFormer/SSM/LinRNN models have different origins and forms.

This work breaks this perception and abstracts existing LinFormer/SSM/LinRNN models into a unified linear attention form, which has the following significance: i) Facilitates understanding

---

*Equal contribution. yuhong.chou@connect.polyu.hk; man.yao@ia.ac.cn
†Corresponding author, guoqi.li@ia.ac.cn

the key designs of existing linear models. Through the unified form, we demonstrate that the main difference between LinFormer/SSM/LinRNN is the hidden state size, how to maintain the hidden state, and how to perform parameter mapping. ii) Links LinFormer/SSM/LinRNN to softmax attention in terms of functionality. The recurrent inference complexity of softmax attention is $\mathcal{O}(n)$, which can also be regarded as the maintenance of a hidden state with infinite size. Linear models with $\mathcal{O}(1)$ inference complexity are hoping to achieve the same functionality as softmax attention using a fixed hidden state. Since we have unified LinFormer/SSM/LinRNN into linear attention in the form of Query, Key, and Value, we can understand and evaluate existing linear models from the view of "Does the linear attention map have the function of softmax attention map?".

To answer this question, we define the necessary conditions for achieving "optimal linear approximation to softmax attention map". First, linear attention needs to satisfy *dynamic memory* and *static approximation* to realize the approximation. The former defines memory ability: linear attention with limited hidden states should be able to store the most important information and forget unimportant ones. The latter defines the modeling ability: a linear attention map should be able to approximate any softmax attention map. According to our theoretical analysis, Query and dynamic decay are necessary conditions for approximation. Thus, linear models such as TransNormer [13], RetNet [14], RWKV-4 [17], LRU [19], HGRN [20], H3 [21], S5 [22], cannot achieve approximation of the softmax attention functions. Second, the Key matrix is not required to achieve approximation, so Mamba [16] and GLA [15] are not optimal parametric approximations.

We then propose the MetaLA module, which can satisfy the necessary conditions for optimal linear approximation to softmax attention. MetaLA makes three enhancements: i) Removes the unnecessary Key matrices; ii) Employs self-augmentation to enhance the token's attention to itself, which avoids attention dilution [12]; iii) Exploits short convolutions to enhance local interactions. We then build a MetaLA Transformer based on MetaLA. Our experiments on associative recall, language modeling, long sequence modeling, and image classification show the effectiveness of MetaLA. Furthermore, we conduct ablation studies to validate the effectiveness of each proposed enhancement in MetaLA. Finally, we discuss two open questions: i) How to further improve linear attention based on the approximation theory introduced in this work? ii) Does the approximation of linear attention to softmax attention imply that it has an upper limit on its capacity?

## 2 Background

For notations in this work, we use bold upper-case letters for matrices (e.g., $\mathbf{Q}$, $\mathbf{K}$), bold lower-case letters for row vectors (e.g., $\mathbf{q}_t$, $\mathbf{k}_t$), and italic upper-case for learnable parameter matrices (e.g., $\boldsymbol{W}_Q$). We generally use the same alphabet to show the rows of a matrix, e.g., $\mathbf{q}_t$ is the $t$-th row of $\mathbf{Q}$.

**Softmax Attention** first calculates an attention map $\mathrm{SoftAttMap}\,(\mathbf{Q},\mathbf{K})$ through $\mathbf{Q}$ (Query), $\mathbf{K}$ (Key), and use the attention map to weight different tokens $\mathbf{V}$ (Value) later:

$$\mathbf{O} = \mathrm{SoftAttMap}\,(\mathbf{Q},\mathbf{K})\,\mathbf{V} = \mathrm{softmax}\left(\frac{\mathbf{Q}\mathbf{K}^\top}{\sqrt{d_k}} \odot \mathbf{M}\right)\mathbf{V} \quad \in \mathcal{R}^{n \times d_v}, \tag{1}$$

$$\mathbf{Q}, \mathbf{K} = \mathbf{X}\boldsymbol{W}_Q, \mathbf{X}\boldsymbol{W}_K \quad \in \mathcal{R}^{n \times d_k}; \quad \mathbf{V} = \mathbf{X}\boldsymbol{W}_V \quad \in \mathcal{R}^{n \times d_v}, \tag{2}$$

where $\boldsymbol{W}_Q, \boldsymbol{W}_K \in \mathcal{R}^{d \times d_k}, \boldsymbol{W}_V \in \mathcal{R}^{d \times d_v}$ are learnable matrices, $n, d, d_k, d_v$ are sequence length, model dimension, Key/Query and Value dimension, respectively. $\mathbf{X} \in \mathcal{R}^{n \times d}$ refers to the input. $\mathbf{M} \in \mathcal{R}^{n \times n}$ is a mask matrix in autoregressive tasks to prevent a token from attending to future tokens. The $t$-th row of $\mathrm{SoftAttMap}\,(\mathbf{Q},\mathbf{K})$ is a probability distribution that represents the attention scores between token $\mathbf{v}_t$ to others. Softmax attention in Eq. (1) enables efficient parallel training, but suffers from $\mathcal{O}(n^2)$ time and memory complexity [9]. It uses the recurrent form during inference:

$$\mathbf{o}_t = \frac{\sum_{s=1}^{t} \exp(\mathbf{q}_t \mathbf{k}_s^\top)\mathbf{v}_s}{\sum_{s=1}^{t} \exp(\mathbf{q}_t \mathbf{k}_s^\top)} \quad \in \mathcal{R}^{1 \times d_v}, \tag{3}$$

$$\mathbf{q}_t, \mathbf{k}_t = \mathbf{x}_t \boldsymbol{W}_Q, \mathbf{x}_t \boldsymbol{W}_K \quad \in \mathcal{R}^{1 \times d_k}; \quad \mathbf{v}_t = \mathbf{x}_t \boldsymbol{W}_V \quad \in \mathcal{R}^{1 \times d_v}. \tag{4}$$

At each time $t$, token mix is computed between query $\mathbf{q}_t$ and all the keys, values before $\mathbf{k}_s, \mathbf{v}_s (s \leq t)$. This "KV cache" results in $\mathcal{O}(n)$ time and memory complexity per token during inference.

**Linear Transformer (LinFormer)** is a substitute for softmax attention, which can be expressed as a linear dot-product of kernel feature maps [9]:

$$\mathbf{o}_t = \frac{\sum_{s=1}^{t} F(\mathbf{q}_t, \mathbf{k}_s)\mathbf{v}_s}{\sum_{s=1}^{t} F(\mathbf{q}_t, \mathbf{k}_s)}, \quad F(\mathbf{q}_t, \mathbf{k}_s) = \phi(\mathbf{q}_t)\phi^\top(\mathbf{k}_s), \tag{5}$$

where $\mathbf{q}_t, \mathbf{k}_t \in \mathcal{R}^{1 \times d_k}$ and $\mathbf{v}_t \in \mathcal{R}^{1 \times d_v}$ are query, key and value at position $t$, which are obtained in the same manner as softmax attention. $F(\cdot)$ is the kernel function usually constrained to be non-negative. $\phi(\cdot)$ is map function applied row-wise to $\mathbf{Q}$ and $\mathbf{K}$. By removing the nonlinear softmax operation, LinFormer enables inference with $\mathcal{O}(1)$ time and memory complexity per token. LinFormer can also be formulated in the following parallel form during training

$$\mathbf{O} = \left(\phi(\mathbf{Q})\phi^\top(\mathbf{K}) \odot \mathbf{M}\right)\mathbf{V} \quad \in \mathcal{R}^{n \times d_v}, \tag{6}$$

which has $\mathcal{O}(n)$ time and memory complexity using chunkwise algorithm. Recent advances in LinFormer mainly focus on training acceleration[12, 13, 23] or improving performance[14].

**State-Space Model (SSM)** come from continuous-time system which maps a 1D function $x(t) \in \mathcal{R}$ to another function $y(t) \in \mathcal{R}$ via a hidden state $\mathbf{h}(t) \in \mathcal{R}^N$. In SSM, the continuous parameters can be discretized using a step size $\mathbf{\Delta}$ and get discrete parameters $\overline{A}, \overline{B}, \overline{C}$. The resulting discrete-time system is used to model sequences $\mathbf{x}, \mathbf{y} \in \mathcal{R}^{1 \times n}$ with elements $x_t, y_t \in \mathcal{R}$ via the recurrent form:

$$\mathbf{h}_t = \overline{A}\mathbf{h}_{t-1} + \overline{B}x_t, \quad y_t = \overline{C}\mathbf{h}_t, \tag{7}$$

in autoregressive inference with $\mathcal{O}(1)$ time and memory complexity per token. The linear time-invariant SSM above can be unrolled and computed using the long convolution with kernel $\mathbf{K}$

$$\mathbf{K} := (\overline{CB}, \overline{CAB}, \cdots, \overline{CA}^{n-1}\overline{B}) \quad \in \mathcal{R}^{1 \times n}, \quad \mathbf{y} = \mathbf{K} * \mathbf{x}, \tag{8}$$

where $*$ represents casual convolution operation [24], which enables parallelizable training utilizing Fast Fourier Transforms, resulting in $\mathcal{O}(n \log n)$ time and $\mathcal{O}(n)$ memory complexity during training. When handling vector sequences $\mathbf{X}, \mathbf{Y} \in \mathcal{R}^{d \times n}$, SSMs are applied individually on the $d$ channels. Typical SSMs (S4D[25], DSS[26], H3[21], S5[22]) employ data-independent structured transition matrix $\overline{\mathbf{A}}$ or special initialization strategies HiPPO[27] to efficiently enhance long-range dependencies. Mamba [16] advances SSMs by introducing data-dependent parameters and designs a hardware-aware parallel algorithm, further improving prior $\mathcal{O}(n \log n)$ into $\mathcal{O}(n)$ time complexity during training.

**Linear RNNs (LinRNN)** Traditional RNNs suffer from slow sequential training, limited capability in modeling long-term dependencies, and difficulty in scaling. To address these issues, LinRNNs eliminate the nonlinearity within the recurrence and employ element-wise product instead of matrix multiplication [11, 28]. Typical LinRNN such as Gated Linear Recurrent Unit (GLRU) [20, 29] is

$$\mathbf{f}_t, \mathbf{i}_t, \mathbf{c}_t = \sigma(\mathbf{x}_t\boldsymbol{W}_f + \boldsymbol{b}_f), \sigma(\mathbf{x}_t\boldsymbol{W}_i + \boldsymbol{b}_i), \phi(\mathbf{x}_t\boldsymbol{W}_c + \boldsymbol{b}_c) \quad \in \mathcal{R}^{1 \times d}, \tag{9}$$

$$\mathbf{h}_t = \mathbf{f}_t \odot \mathbf{h}_{t-1} + \mathbf{i}_t \odot \mathbf{c}_t, \in \mathcal{R}^{1 \times d}, \tag{10}$$

where $\mathbf{x}_t, \mathbf{h}_t$ denote input and output, $\mathbf{f}_t, \mathbf{i}_t$ are forget and input gates as in traditional RNNs, $\odot$ is element-wise multiplication. Linear RNNs have $\mathcal{O}(1)$ time and memory complexity per token during inference. Since Eq. (10) removes nonlinearity, it enables parallelized training using parallel scan[11], with only $\mathcal{O}(n)$ time and memory complexity. Recent works have made effort to explore effective recurrence (LRU [19], RWKV [17]) or gating mechanisms (HGRN [20], Griffin [18]).

## 3 General Form of LinFormer/SSM/LinRNN Mechanisms

Observing Eq. (3), Eq. (5), Eq. (7), and Eq. (10), we find that their recurrent forms during inference can all be understood from the view of maintaining hidden states. Softmax attention maintains an unlimited hidden state (KV cache). By contrast, LinFormer/SSM/LinRNN have limited hidden states: linear attention with $\phi^\top(\mathbf{k}_t)\mathbf{v}_t \in \mathcal{R}^{d_k \times d_v}$, SSM with $\mathbf{h}_t \in \mathcal{R}^{N \times d}$, linear RNNs with $\mathbf{h}_t \in \mathcal{R}^{1 \times d}$, where $d_k > N > 1$ in usual. Inspired by this fact, we unify LinFormer/SSM/LinRNN mechanisms in the form of linear attention, formally containing Query, Key, and Value matrices (see Fig. 1).

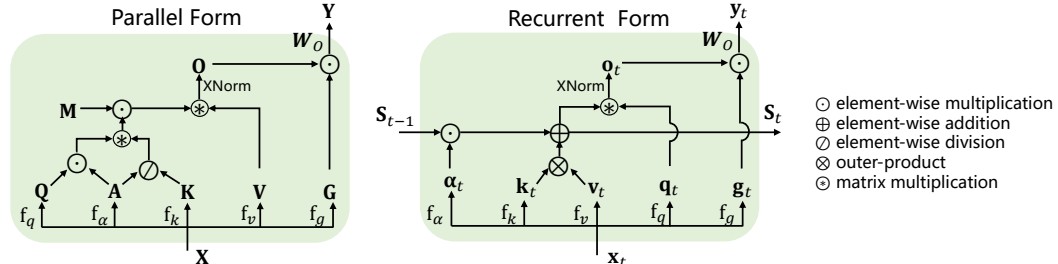

Figure 1: **General Form of LinFormer/SSM/LinRNN Mechanisms.** The general form equips with two modes of parallel and recurrent computation which enjoys both training and inference efficiency.

**General Recurrent Form of LinFormer/SSM/LinRNN** is:

$$\mathbf{q}_t = \mathrm{f}_q(\mathbf{x}_t, \theta_q), \mathbf{k}_t = \mathrm{f}_k(\mathbf{x}_t, \theta_k), \alpha_t = \mathrm{f}_\alpha(\mathbf{x}_t, \theta_\alpha) \quad \in \mathcal{R}^{1 \times d_k}, \tag{11}$$

$$\mathbf{v}_t = \mathrm{f}_v(\mathbf{x}_t, \theta_v), \mathbf{g}_t = \mathrm{f}_g(\mathbf{x}_t, \theta_g) \quad \in \mathcal{R}^{1 \times d_v}, \tag{12}$$

$$\mathbf{S}_t^h = \mathrm{diag}(\alpha_t^h)\mathbf{S}_{t-1}^h + (\mathbf{k}_t^h)^\top \mathbf{v}_t^h \quad \in \mathcal{R}^{d_k' \times d_v'}, \tag{13}$$

$$\mathbf{o}_t = \mathrm{XNorm}(\mathrm{concat}[\mathbf{q}_t^1 \mathbf{S}_t^1, \mathbf{q}_t^2 \mathbf{S}_t^2, \cdots, \mathbf{q}_t^H \mathbf{S}_t^H]) \quad \in \mathcal{R}^{1 \times d_v}, \tag{14}$$

$$\mathbf{y}_t = (\mathbf{o}_t \odot \mathbf{g}_t)\boldsymbol{W}_O \quad \in \mathcal{R}^{1 \times d}, \tag{15}$$

where $\mathbf{x}_t \in \mathcal{R}^{1 \times d}$ is the $t$-th input. $\mathbf{q}_t, \mathbf{k}_t, \mathbf{v}_t, \alpha_t, \mathbf{g}_t, \mathbf{S}_t$ are query, key, value, decay, output gate, hidden state respectively. $\mathrm{f}_{q/k/\alpha}$ are functions that map $\mathbf{x}_t$ from $\mathcal{R}^{1 \times d}$ to $\mathcal{R}^{1 \times d_k}$, $\theta_{q/k/\alpha}$ are the corresponding parameters to be trained. Similarly, $\mathrm{f}_{v/g}$ map $\mathbf{x}_t$ from $\mathcal{R}^{1 \times d}$ to $\mathcal{R}^{1 \times d_v}$ and $\theta_{v/g}$ are trainable parameters. In Eq. (13), $\mathbf{q}_t, \alpha_t, \mathbf{k}_t, \mathbf{v}_t$ are divided into $H$ partitions (heads), where $d_{k/v}' = \frac{d_{k/v}}{H}$, and $h = 1, \cdots, H$ is the index of heads. Each head maintains a hidden state $\mathbf{S}_t^h$. The diagonal matrix $\mathrm{diag}(\alpha_t^h)$ denotes the decay of past state. $\mathbf{k}_t^h$ represents the acceptance for the input token $\mathbf{v}_t^h$. The hidden states are 2D matrix once $d_k' \neq 1$. In Eq. (14), to turn back to 1D shape, the $\mathbf{q}_t^h$ operation is necessary as a dot-product with $\mathbf{S}_t^h$, then the concat and normalization operations are followed. XNorm denotes any kinds of normalization. In Eq. (15), a gate machanism is optional for $\mathbf{o}_t$ while the dimension should be projected back to $d$ from $d_v$ through $\boldsymbol{W}_O \in \mathcal{R}^{d_v \times d}$.

From a functional view, Eq. (13) represents the update process of the hidden state, which contains historical information on keys and values. Eq. (14) represents query and weighted sum operations to derive the attention output. Eq. (15) represents gate and projection operations to get the final output.

**General Parallel Form of LinFormer/SSM/LinRNN** can be written as follow:

$$\mathbf{O} = \mathrm{LinAttMap}\,(\mathbf{Q}, \mathbf{K})\,\mathbf{V} = \left(\left(\left(\mathbf{Q} \odot \mathbf{A}\right)\left(\frac{\mathbf{K}}{\mathbf{A}}\right)^\top\right) \odot \mathbf{M}\right)\mathbf{V}, \tag{16}$$

$$(\mathbf{Q}/\mathbf{K}/\mathbf{V})_{t,:} = (\mathbf{q}/\mathbf{k}/\mathbf{v})_t, \quad \mathbf{A}_{t,:} = \prod_{j=1}^{t} \alpha_j, \quad \mathbf{M}_{i,j} = \begin{cases} 1, i \leq j. \\ 0, i > j. \end{cases} \tag{17}$$

$\frac{\mathbf{K}}{\mathbf{A}}$ denotes element-wise division, $(\mathbf{Q})_{t,:}$ is the $t$-th row of $\mathbf{Q}$, and $\mathrm{LinAttMap}\,(\mathbf{Q}, \mathbf{K})$ is the attention map. Each element in the attention map matrix is as follows (heads are omitted for simplicity):

$$\mathrm{LinAttMap}\,(\mathbf{Q}, \mathbf{K})_{t,s} = \begin{cases} \mathbf{q}_t \cdot \left(\left(\prod_{j=s+1}^{t} \alpha_j\right) \odot \mathbf{k}_s\right)^\top, s \leq t. \\ 0, s > t. \end{cases} \tag{18}$$

As shown in Tab. 1, the main differences between various linear models are parameter functions $\mathrm{f}_{q/k/v/\alpha/g}$ and dimension settings $d_k, d_v, H$. We give details in appendix A1 on how to derive LinFormers/SSMs/linRNNs from our unified form, which is termed as "Linear Attention (LinAtt)".

LinFormer/SSM/LinRNN models have different origins, so they have different **optimization perspectives** and various **hidden state sizes**: i) *LinFormer* originate from approximation of vanilla softmax attention. They focus on designing better kernel function $\phi$, i.e., to optimize $\mathrm{f}_q, \mathrm{f}_k$; They have

Table 1: **From our general form to existing linear models** ($*$ indicates the bias term is omitted).

| Models | LinFormer | | LinRNN | | SSM | |
|---|---|---|---|---|---|---|
| | GLA[15] | TrNorm[12] | GLRU[20] | RWKV-4[17] | Mamba[16] | S5[22] |
| $f_q(\mathbf{x}_t, \theta_q)$ | $\mathbf{x}_t \boldsymbol{W}_Q$ | $\phi(\mathbf{x}_t \boldsymbol{W}_Q)$ | $\mathbf{1}$ | $\mathbf{1}$ | $\mathbf{x}_t \boldsymbol{W}_C$ | $\mathbf{1}$ |
| $f_k(\mathbf{x}_t, \theta_k)$ | $\mathbf{x}_t \boldsymbol{W}_K$ | $\phi(\mathbf{x}_t \boldsymbol{W}_K)$ | $\sigma(\mathbf{x}_t \boldsymbol{W}_i)^*$ | $\exp(\mathbf{x}_t \boldsymbol{W}_K)$ | $\Delta_t(\mathbf{x}_t \boldsymbol{W}_b)$ | $\mathbf{1}$ |
| $f_v(\mathbf{x}_t, \theta_v)$ | $\mathbf{x}_t \boldsymbol{W}_V$ | $\mathbf{x}_t \boldsymbol{W}_V$ | $\phi(\mathbf{x}_t \boldsymbol{W}_c)^*$ | $\mathbf{x}_t \boldsymbol{W}_V$ | $\mathbf{x}_t$ | $\mathbf{x}_t \overline{\boldsymbol{B}}$ |
| $f_\alpha(\mathbf{x}_t, \theta_\alpha)$ | $\sigma(\mathbf{x}_t \boldsymbol{W}_\alpha^1 \boldsymbol{W}_\alpha^2)^*$ | $\lambda \exp(j\theta)$ | $\sigma(\mathbf{x}_t \boldsymbol{W}_f)^*$ | $\exp(-\boldsymbol{W})$ | $\exp(\Delta_t \boldsymbol{A})$ | $\exp(\Delta \boldsymbol{A})$ |
| $f_g(\mathbf{x}_t, \theta_g)$ | $\sigma(\mathbf{x}_t \boldsymbol{W}_r)^*$ | $\mathbf{x}_t \boldsymbol{W}_U$ | $\phi(\mathbf{x}_t \boldsymbol{W}_g)^*$ | $\sigma(\mathbf{x}_t \boldsymbol{W}_r)$ | $\mathbf{1}$ | $\mathbf{1}$ |
| Dimension | $d_k = d/2$ $d_v = d$ | $d_k = d_v = d$ | $d_k = d_v = d = H$ | | $d_v = d = H$ $d'_k = N$ | $d_v = Nd = H$ $d'_k = 1$ |

relatively large matrix hidden state whose size ($d_v d_k / H$) is mainly correlated to model dimension $d$. ii) *SSM* originate from state space equations. So they focus on how to better maintain the hidden state and optimize $f_\alpha$; They have a matrix hidden state of moderate size ($d_v N$), which is correlated to the fixed expansion $N$. iii) *LinRNN* originate from removing nonlinearity in the recurrence of vanilla RNN. So they focus on designing better forget/input/output gates, i.e., to optimize $f_\alpha, f_k, f_g$; They have 1D vector hidden state whose size ($d_v$) is relatively small. Despite these differences, they all try to design better parameter functions $f_{q/k/v/\alpha/g}$ and maintain a limited hidden state $\mathbf{S}_t$.

# 4 Optimal Linear Approximation to the Softmax Attention Map

We here discuss the optimal approximation of LinAttMap to SoftAttMap based on its general form. The function of softmax attention is two-fold: i) *Memorizing information,* all the current and historical information can be stored in KV cache; ii) *Modeling relationships,* softmax attention can calculate arbitrary attention scores of stored information. Unfortunately, such a powerfully expressive attention map generated by softmax attention requires infinite hidden states. By contrast, linear attention expects to exploit limited hidden states to achieve the same functionality as softmax attention.

Some existing linear models, such as Performer[30], RFA[31], etc., optimize the model with the goal of approximating the value of SoftAttMap. In contrast, this work investigates the functional approximation of SoftAttMap, which is the basis for value approximation. Specifically, we here attempt to answer two key questions: i) Can linear attention realize the function of softmax attention? ii) If it can be achieved, what kind of linear attention approximation is better? To achieve this goal, we first give the definition of necessary conditions of optimal linear approximation. Then we categorize the existing linear models based on the conditions of the optimal linear approximation.

**Definition 4.1. Necessary Conditions of Optimal Linear Approximation to Softmax Attention Map.** A function $f(\mathbf{x}_t, \mathbf{x}_s | \theta) : \mathcal{R}^{1 \times d} \times \mathcal{R}^{1 \times d} \to \mathcal{R}$, used to compute attention score between any $\mathbf{x}_t$ and $\mathbf{x}_s$ (tokens), with parameters $\theta$, is an optimal linear approximation to softmax attention map if it satisfies: i) **Linear complexity**. Attention map can be computed in linear time, i.e., $\mathcal{O}(n)$ space and time complexity during training and $\mathcal{O}(1)$ space and time complexity during inference. ii) **Dynamic memory ability**. When handling inputs sequentially, $f(\mathbf{x}_t, \mathbf{x}_s | \theta)$ with limited hidden states should be able to store the most important information adaptively while forgetting unimportant ones. iii) **Static approximation ability**: For an arbitrarily given softmax attention map $\mathbf{P}$ with scores $p_{ts}$, there must exists bounded $\theta$ such that $f(\mathbf{x}_t, \mathbf{x}_s | \theta) = p_{ts}, \forall t, s = 1, \cdots, n$. iv) **Least parameter approximation**: On the premise that the first three conditions are met, use as few parameters as possible to achieve approximation to softmax attention map.

In definition 4.1, Condition 0 (C0) underlines computational and memory efficiency. Conditions 1 (C1) and 2 (C2) consider *memory* and *modeling* ability of linear attention. Due to limited state size $d$, linear attention can only memorize the history of most important $d$ tokens without information loss and precisely model arbitrary attention map of those $d$ tokens. Condition 3 (C3) is our expectation to seek the least parameters on the premise that previous three conditions are met.

**Theoretical Analysis for Optimal Linear Approximation.** For the C1 condition, suppose the information about $\mathbf{v}_{t_1}, \ldots, \mathbf{v}_{t_{d_k}}$ is successfully stored in $\mathbf{S}_t$ ($t_1, \ldots, t_{d_k} \leq t$), we will check whether the model can substitute unimportant $\mathbf{v}_{t_1}$ when the new important input $\mathbf{v}_{t+1}$ arrives.

For the C2 condition, Eq. (18) illustrates the LinAttMap only relate to $\mathbf{q}_t = f_q(\mathbf{x}_t, \theta_q), \mathbf{k}_t = f_k(\mathbf{x}_t, \theta_k), \alpha_t = f_\alpha(\mathbf{x}_t, \theta_\alpha)$. Denote decay $\mathbf{\Lambda}_t = \text{diag}(\alpha_t)$. Assuming the inputs are good enough

Table 2: **A review of existing linear models.** According to definition 4.1, existing linear models all have some deficiencies: i) Models without dynamic decay $\Lambda_t$ have no ability to memorize dynamically (not satisfying C1); ii) LinRNNs lack the selection ability brought by $\mathbf{Q}$ (not satisfying C2), and the approximation ability is poor owing to the small hidden state; iii) Models with $\mathbf{K}$ have redundant parameters (not satisfying C3), which probably leads to higher learning difficulty.

| | | | C0 | C1 | C2 | C3 | Models |
|---|---|---|---|---|---|---|---|
| Softmax Attention | | | ✗ | ✓ | ✓ | - | Transformer [1], GPT [2], Llama [3] |
| Linear Attention | Three Parameter Groups | $\mathbf{Q}, \mathbf{K}, \mathbf{\Lambda}$ | ✓ | ✗ | ✗ | ✗ | RetNet [14], TransNormer [12], S4D [25], H3 [21], DSS [26] |
| | | $\mathbf{Q}, \mathbf{K}, \mathbf{\Lambda}_t$ | ✓ | ✓ | ✓ | ✗ | GLA [15], Mamba [16] |
| | Two Parameter Groups | $\mathbf{Q}, \mathbf{K}$ | ✓ | ✗ | ✓ | ✗ | linear Transformer [9], RFA [31], Performer [30], cosFormer [32] |
| | | $\mathbf{K}, \mathbf{\Lambda}$ | ✓ | ✗ | ✗ | ✗ | RWKV-4 [17] |
| | | $\mathbf{K}, \mathbf{\Lambda}_t$ | ✓ | ✓ | ✗ | ✗ | GLRU [20] |
| | | $\mathbf{Q}, \mathbf{\Lambda}$ | ✓ | ✗ | ✗ | ✗ | - |
| | | $\mathbf{Q}, \mathbf{\Lambda}_t$ | ✓ | ✓ | ✓ | ✓ | **MetaLA (This Work)** |
| | One Parameter Group | $\mathbf{Q}$ or $\mathbf{K}$ | ✓ | ✗ | ✗ | ✗ | - |
| | | $\mathbf{\Lambda}$ | ✓ | ✗ | ✗ | ✗ | LRU [19], S5 [22] |
| | | $\mathbf{\Lambda}_t$ | ✓ | ✓ | ✗ | ✗ | HGRN [20], Griffin [18], T-RNN [33] |

and the functions $(f_q, f_k, f_\alpha)$ are expressive enough, we can shift from solving $(\theta_q, \theta_k, \theta_\alpha)$ to solving $(\mathbf{Q}, \mathbf{K}, \mathbf{\Lambda}_t)$. We focus on approximating the attention scores between stored tokens, and the problem is simplified via: i) setting query dimension $d_k = 1$; ii) considering only a given time $t$ and its attention distribution $\mathbf{p}_t = [p_{ts}, s = 1, \ldots, t] \in \mathcal{R}^{1 \times t}$. Then, C2 is proved by the following equations holding with bounded parameters, as a foundation conclusion:

$$f(\mathbf{x}_t, \mathbf{x}_s | \mathbf{Q}, \mathbf{K}, \mathbf{\Lambda}_t) = q_t \big( \prod_{j=s+1}^{t} \alpha_j \big) k_s = p_{ts}, \forall s = 1, \ldots, t, \qquad (19)$$

$$\text{s.t.} \quad |q_s| \le C_q, |k_s| \le C_k, \alpha_s \in [0, 1], \forall s = 1, \ldots, t. \qquad (20)$$

For bounded inputs $\mathbf{X}$, bounded parameters $(\theta_q, \theta_k, \theta_\alpha)$ are equivalent to bounded $(\mathbf{Q}, \mathbf{K}, \mathbf{\Lambda}_t)$. Afterwards we will generalize to vector version with $d_k > 1$ and consider distribution of all time $(\mathbf{p}_t, t = 1, \ldots, d_k)$. This is done by viewing $\mathbf{q}_t$ as a channel selector.

For the C3 condition, least parameters mean the fewest parameter groups $(\mathbf{Q}, \mathbf{K}, \mathbf{\Lambda}_t)$ when $d, d_k, d_v$ are fixed. Due to space constraints, the detailed analysis in this Section is provided in appendix A2.

**Conclusions of Optimal Linear Approximation Analysis.** i) *Linear approximation.* The necessary conditions (C1 and C2) for LinAttMap to achieve approximation to SoftAttMap is that its implementation must include $\mathbf{Q}$ and dynamic decay $\mathbf{\Lambda}_t$. Both $(\mathbf{Q}, \mathbf{K}, \mathbf{\Lambda}_t)$ and $(\mathbf{Q}, \mathbf{\Lambda}_t)$ can achieve approximation. ii) *Least parameter approximation.* $(\mathbf{Q}, \mathbf{\Lambda}_t)$ has fewer parameters (i.e., $\mathbf{K}$ is not necessary), if the model dimensions are fixed. iii) *Function of dynamic decay.* $\mathbf{\Lambda}_t$ is the key to achieve dynamic memory. iv) *Function of Query.* $\mathbf{Q}$ can be seen as a channel selector which selects several channels of Hadamard product of $\mathbf{\Lambda}_t$ and $\mathbf{K}$ to approximate attention map.

In Tab. 2, we review some existing linear models and judge whether they meet the necessary conditions for optimal approximation. Linear attentions can be classified into three types based on the parameter groups: i) Using $(\mathbf{Q}, \mathbf{K}, \mathbf{\Lambda}_t)$ all together, ii) Exploiting $(\mathbf{Q}, \mathbf{K})$, $(\mathbf{Q}, \mathbf{\Lambda}_t)$ or $(\mathbf{K}, \mathbf{\Lambda}_t)$, iii) Employing only one of $\mathbf{Q}, \mathbf{K}, \mathbf{\Lambda}_t$. Considering decay can be either dynamic or fixed, here we use subscript $t$ to distinguish, i.e., $\mathbf{\Lambda}/\mathbf{\Lambda}_t$ denote fixed/dynamic decay. According to definition 4.1, they have different degrees of deficiencies: i) Models without dynamic decay such as RetNet[14], TransNormer[12], RFA[31], cannot memorize dynamically; ii) LinRNNs such as RWKV-4[17], HGRN[20], Griffin[18]

Table 3: **From general recurrent linear form to our MetaLA.**

| Models | $f_q(\mathbf{x}_t, \theta_q)$ | $f_k(\mathbf{x}_t, \theta_k)$ | $f_v(\mathbf{x}_t, \theta_v)$ | $f_\alpha(\mathbf{x}_t, \theta_\alpha)$ | $f_g(\mathbf{x}_t, \theta_g)$ | Dimension |
|---|---|---|---|---|---|---|
| MetaLA | $\mathbf{x}_t \boldsymbol{W}_Q$ | $1 - \alpha_t$ | $\mathbf{x}_t \boldsymbol{W}_V$ | $\sigma(\mathbf{x}_t \boldsymbol{W}_\alpha)$ | $\phi(\mathbf{x}_t \boldsymbol{W}_G + \boldsymbol{b}_G)$ | $d_k = d/2, d_v = d$ |

lack the selection ability brought by $\mathbf{Q}$ and the approximation ability is poor due to the small hidden state; iii) Models with $\mathbf{K}$ such as Mamba[16], GLA[15] have redundant parameters, which probably leads to higher learning difficulty. Thus, none of the existing linear models meet all C1/C2/C3 conditions. These analyses also inspire us that ignoring the functional approximation of softmax attention does not enable the approximation of softmax attention values.

## 5 MetaLA Transformer

Transformer is stacked by a series of Encoder/Decoder blocks. Generally, each block is composed of two modules in sequence: token mixer and channel mixer [34, 35]. Softmax attention plays the role of the token mixer. In this work, we follow the Transformer architecture as a whole but use our proposed MetaLA module as the token mixer. Due to space constraints, the architecture of MetaLA Transformer is given in detail in appendix A3. Here we focus on describing the three enhancements of MetaLA relative to the general linear attention in Sec. 3 (see Fig. 2): i) The Key matrix is not used, which is based on our theoretical analysis. ii) Self-augmentation and iii) Short convolution are two other optional techniques to further enhance the modeling ability of our model.

i) *The Key matrix is not used.* We exploit $1 - \alpha_t$ to replace $\mathbf{k}_t$, based on theoretical analysis in Sec. 4 and appendix A2, i.e., dynamic decay $\boldsymbol{\Lambda}_t$ is the key mechanism to achieve dynamic memory and static approximation, and $\mathbf{K}$ is not required. As shown in Tab. 3, compared with Eq. (13), the main improvement is:

$$\mathbf{S}_t^h = \text{diag}(\alpha_t^h)\mathbf{S}_{t-1}^h + (1 - \alpha_t^h)^\mathsf{T}\mathbf{v}_t \quad \in \mathcal{R}^{d_k' \times d_v'}, \tag{21}$$

which can be trained in a parallel form in Eq. (16). The only difference is that $\mathbf{K}$ is replaced by $\mathbf{B}$ and $\mathbf{B}_{t,:} = 1 - \alpha_t$. With usage of $\boldsymbol{\Lambda}_t$ and $\mathbf{Q}$, MetaLA can achieve linear approximation to SoftAttMap. Without

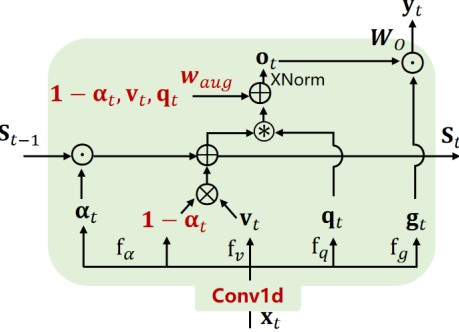

Figure 2: **Recurrent form of MetaLA.** We mark all three enhancements in red.

usage of $\mathbf{K}$ ($\boldsymbol{W}_K$), we can allocate more parameters and utilize full-rank matrix $\boldsymbol{W}_\alpha$ to produce dynamic decay rather than low-rank matrix used by GLA, such that we do not sacrifice expression capacity of $f_\alpha$ and approximation performance of MetaLA.

ii) *Self-augmentation* can enhance a token's attention to itself, avoiding attention dilution [12]:

$$\mathbf{o}_t^h = \mathbf{q}_t^h \mathbf{S}_t^h + \sigma_{\text{aug}}\big(\mathbf{q}_t^h(\boldsymbol{w}_{\text{aug}}^h \odot (1 - \alpha_t^h))^\mathsf{T}\mathbf{v}_t\big) \quad \in \mathcal{R}^{1 \times d_v'}. \tag{22}$$

Without changing the hidden state $\mathbf{S}_t^h$ in Eq. (21), the proposed self-augmentation (the second term on the right side of the equation) is only added on the output process, with a learnable parameter $\boldsymbol{w}_{\text{aug}} \in \mathcal{R}^{1 \times d_k}$. The proposed design has two advantages (more analysis in appendix A3.2): First, it maintains parallel computing; Second, it augments the information of current token itself and does not affect future output through inner state.

iii) *Short convolution.* An additional short convolution can be inserted before entering the MetaLA layer to enhance local interaction further, motivated by Mamba [16] and Griffin [18].

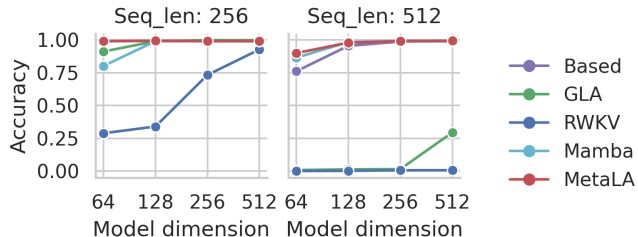

Figure 3: **Accuracy (%) on the synthetic MQAR task.**

## 6 Experiments

We conduct a comprehensive evaluation of MetaLA to validate its

Table 4: **Performance Comparison on SuperGLUE.** PS: parameter size (billion). T: tokens (billion). [†] means the results reported by [20]. For baselines that need to be compared, if they do not have public checkpoints, we train and test them under identical conditions with MetaLA. MetaLA$_a$: MetaLA with tied embedding trained using 100B tokens. MetaLA$_b$: MetaLA trained with 300B tokens.

| Models | PS | T | WSC | WIC | RTE | CB | MULTIRC | BOOLQ | COPA | AVG |
|---|---|---|---|---|---|---|---|---|---|---|
| Pythia | 0.41 | 15 | 36.54 | 50.00 | 52.35 | 39.29 | 0.31 | 61.99 | 62.00 | 43.21 |
| Mamba | 0.37 | 15 | 36.54 | 50.31 | 52.71 | 42.86 | 2.52 | 58.78 | 64.00 | 43.96 |
| GLA | 0.36 | 15 | 36.54 | 49.84 | 53.07 | 41.07 | 0.42 | 53.49 | 66.00 | 42.92 |
| MetaLA | 0.36 | 15 | 36.54 | 50.00 | 52.71 | 42.86 | 0.31 | 58.96 | 67.00 | **44.05** |
| Pythia[†] | 1.4 | 300 | 36.54 | 50.00 | 53.07 | 35.71 | 0.94 | 60.73 | 72.00 | 44.14 |
| HGRN[†] | 1 | 100 | 40.38 | 50.78 | 53.43 | 42.86 | 3.04 | 58.69 | 70.00 | 45.60 |
| Mamba | 1.4 | 100 | 39.42 | 50.94 | 55.23 | 26.79 | 1.15 | 53.27 | 73.00 | 42.83 |
| RetNet[‡] | 1.3 | 100 | 36.54 | 50.00 | 52.71 | 46.43 | 2.52 | 60.21 | 68.00 | 45.20 |
| GLA[‡] | 1.3 | 100 | 36.54 | 50.16 | 53.07 | 37.50 | 0.31 | 61.04 | 69.00 | 43.95 |
| MetaLA$_a$ | 1.3 | 100 | 49.04 | 51.25 | 55.60 | 37.50 | 1.78 | 55.50 | 70.00 | **45.81** |
| MetaLA$_b$ | 1.4 | 300 | 62.50 | 51.88 | 49.10 | 48.21 | 1.57 | 56.27 | 75.00 | **49.22** |

Table 5: **Performance Comparison on Commonsense Reasoning.** [‡] indicates testing using open-source checkpoints. HS: HellaSwag. WG: WinoGrande. OBQA: OpenbookQA.

| Models | PS | T | LOGIQA | WSC273 | BOOLQ | PIQA | HS | WG | ARC-c | OBQA | AVG |
|---|---|---|---|---|---|---|---|---|---|---|---|
| Pythia | 0.41 | 15 | 21.81 | 57.51 | 61.99 | 63.66 | 33.15 | 51.78 | 22.78 | 28.60 | **42.66** |
| Mamba | 0.37 | 15 | 20.43 | 56.78 | 58.78 | 64.80 | 33.98 | 49.80 | 22.87 | 29.20 | 42.08 |
| GLA | 0.36 | 15 | 23.04 | 56.78 | 53.49 | 63.55 | 32.00 | 52.10 | 22.78 | 27.40 | 41.39 |
| MetaLA | 0.36 | 15 | 22.43 | 58.24 | 58.96 | 63.82 | 32.18 | 53.12 | 23.38 | 28.00 | 42.52 |
| Pythia[‡] | 1.4 | 300 | 21.35 | 72.89 | 63.12 | 70.89 | 51.98 | 56.99 | 28.41 | 33.20 | 49.85 |
| HGRN[‡] | 1 | 100 | 22.43 | 58.97 | 58.75 | 71.00 | 48.05 | 51.14 | 28.07 | 31.80 | 46.28 |
| Mamba | 1.4 | 100 | 22.73 | 68.50 | 53.27 | 71.44 | 48.63 | 53.59 | 29.01 | 31.80 | 47.37 |
| RetNet[‡] | 1.3 | 100 | 22.73 | 63.74 | 60.21 | 69.53 | 48.39 | 53.28 | 26.19 | 30.80 | 46.86 |
| GLA[‡] | 1.3 | 100 | 21.81 | 63.00 | 61.04 | 70.08 | 48.00 | 51.93 | 28.33 | 31.40 | 46.95 |
| MetaLA$_a$ | 1.3 | 100 | 21.81 | 65.93 | 55.50 | 70.02 | 47.32 | 55.01 | 27.47 | 33.00 | 47.01 |
| MetaLA$_b$ | 1.4 | 300 | 21.35 | 73.63 | 56.27 | 72.25 | 53.58 | 58.17 | 30.03 | 34.60 | **49.99** |

capabilities as a foundation model. i) MQAR [36]. Performance on the Multi-Query Associative Recall (MQAR) task is closely linked to language modeling and can also imply the effectiveness of our theory in modeling hidden states and retrieving information. ii) Autoregressive language modeling on the Pile [37] dataset and evaluation on Common-Sense Reasoning and SuperGLUE [38] zero-shot benchmarks are conducted. iii) LRA [39]. We execute experiments on the Long Range Arena (LRA) benchmark [39] to investigate MetaLA's ability in long sequence modeling. iv) ImageNet [40]. Generalization ability in visual tasks. Due to space constraints, we put some additional experiments in appendix A5, including: v) Scalability. We extend MetaLA to a 3B parameter scale and a 300B data scale for preliminary validation. vi) Retrieval and long context abilities. We evaluated MetaLA's retrieval performance on the MAD tasks [41], and its effectiveness in handling long contexts on the Needle in a Haystack task [42]. vii) Training efficiency. We provide comparative results on training throughput and GPU memory usage across various models. Detailed experimental setup and further discussion are given in appendix A4.

**Associative Recall.** The synthetic MQAR task [36] is exploited to evaluate MetaLA's memory ability. In the task, given multiple queries, the model must recall the corresponding key-value pairs before. We follow default settings in [36] to generate datasets. Fig. 3 shows that MetaLA outperforms other linear models, which have three parameter groups (Mamba [16], GLA [15], Based [43]) or fixed decay (RWKV-4 [17]), well supporting our theoretical analysis and module design. The attention baseline achieves optimal results ($> 99.0$) under both conditions. The additional experiments in appendix A5 show that MetaLA outperforms Mamba on more challenging settings.

**Language Modeling.** We train two scales of MetaLA: 360M/1.4B on the Pile dataset. For baselines of 360M MetaLA, we train them from scratch aligned with our configurations. For the 1.3B MetaLA,

Table 7: **Performances Comparison on the Long Range Arena.** We cite baselines from HGRN [20].

| Method | LitsOps | Text | Retrieval | Image | Pathfinder | Path-X | AVG. |
|---|---|---|---|---|---|---|---|
| Transformer [1] | 38.37 | 61.95 | 80.69 | 40.57 | 65.26 | - | 47.81 |
| S4 [10] | 59.60 | 86.82 | 90.90 | 88.65 | 94.20 | 96.35 | 86.09 |
| DSS-softmax [26] | 60.60 | 84.80 | 87.80 | 85.70 | 84.60 | 87.80 | 81.88 |
| TNN [46] | 61.04 | 87.90 | 90.97 | 88.24 | 93.00 | 96.10 | 86.21 |
| S5 [22] | 62.15 | 89.31 | 91.40 | 88.00 | 95.33 | 98.56 | 87.46 |
| Mega [47] | 63.14 | 90.43 | 91.25 | 90.44 | 96.01 | 97.98 | **88.21** |
| SGConv [48] | 61.45 | 89.20 | 91.11 | 87.97 | 95.46 | 97.83 | 87.17 |
| LRU [19] | 60.20 | 89.40 | 89.90 | 89.00 | 95.10 | 94.20 | 86.30 |
| HGRN [20] | 59.95 | 88.14 | 94.23 | 88.69 | 92.92 | 97.50 | 86.91 |
| Mamba [16] | 38.02 | 82.98 | 72.14 | 69.82 | 69.26 | 67.32 | 66.59 |
| MetaLA(ours) | 59.34 | 89.27 | 91.28 | 91.88 | 91.66 | 96.57 | 86.67 |

Table 8: **Ablation studies.** Ablation study results on the 360M model trained for 15B tokens. We compared the model variants on zero-shot experiments of the Comparison on Commonsense Reasoning benchmark. HS: HellaSwag. WG: WinoGrande. OBQA: OpenbookQA.

| Models | LOGIQA | WSC273 | BOOLQ | PIQA | HS | WG | ARC-c | OBQA | AVG |
|---|---|---|---|---|---|---|---|---|---|
| MetaLA | 22.43 | 58.24 | 58.96 | 63.82 | 32.18 | 53.12 | 23.38 | 28.00 | **42.52** |
| MetaLA w/o selfaug | 21.81 | 58.61 | 57.52 | 64.47 | 32.56 | 49.41 | 23.89 | 29.00 | 42.16 |
| MetaLA w/o conv | 22.58 | 51.65 | 49.36 | 52.07 | 25.82 | 51.22 | 26.54 | 28.80 | 38.51 |
| MetaLA w/ key | 21.20 | 57.88 | 49.11 | 63.00 | 32.99 | 50.99 | 23.63 | 27.60 | 40.80 |

we compare it with publicly available models [14, 15, 16, 20, 44]. We implement all the pre-train experiments with GPT-Neox [45]. The zero-shot evaluation results on SuperGLUE and Commensense Reasoning benchmarks are reported in Tab. 4 and Tab. 5. Specifically, compared to the LinRNN model HGRN [20], MetaLA expands hidden state dimensions and uses Query matrix; Compared to LinFormer model RetNet [14] with fixed decay, MetaLA uses dynamic decay; Compared to SSMs like Mamba [16] and LinFormer with dynamic decay GLA [15], MetaLA omits the Key matrix in computation. Results indicate that MetaLA has better performance than these linear models and the Transformer-based Pythia [44]. See appendix A5 for more task results.

**Long Sequence Modeling.** LRA is used to evaluate the model's ability in long sequence modeling. We compare MetaLA with Transformer, linear foundation models, and models specifically designed for long sequence modeling. Tab. 7 shows that MetaLA achieves comparable results with SOTA linear models, demonstrating that our model effectively preserves the ability to model long sequences.

Table 6: **Results on ImageNet-1k.**

| Model | Acc | PS (M) | Acc | PS (M) |
|---|---|---|---|---|
| Deit | 72.20 | 5.7 | 79.90 | 22.0 |
| HGRN | 74.40 | 6.1 | 80.09 | 23.7 |
| GLA | 72.47 | 6.1 | 79.23 | 23.5 |
| Mamba | 73.39 | 6.1 | 79.60 | 23.7 |
| MetaLA | **75.33** | 6.1 | **80.14** | 23.7 |

**Image Classification.** We compare MetaLA with Deit [49] (Transformer), HGRN [20] (LinRNN), GLA [15] (LinFormer) and Mamba [16] (SSM) on ImageNet. As shown in Tab. 6, MetaLA performs better than other typical linear models at both scales of 6M and 23M.

**Ablation Studies.** We conduct ablation studies on the 360M model trained with 15B tokens and compare the results in zero-shot experiments. First, restoring the Key matrix in linear attention does not improve performance while increasing parameters, supporting our theoretical result that **K** is not necessary for approximation, and its functional role can be replaced by dynamic decay. Second, the ablations of self-augmentation and short convolution demonstrate the effectiveness of our model design, i.e., enhance tokens' own attention and local interactions.

## 7 Conclusion and Discussion

**Conclusion.** We unify LinFormer/SSM/LinRNN models into the form of linear attention with Query, Key, Vaule matrices, and then analyze whether they can achieve the optimal approximation to the softmax attention function. Theoretical analysis shows that the existing LinFormer/SSM/LinRNN

*cannot* achieve optimal approximation. Consequently, we propose the MetaLA architecture, which can achieve functional approximation of softmax attention with least parameters. The performance on various types of tasks verifies the effectiveness of MetaLA.

**Discussion and Limitation.** Here, we discuss two key questions about approximation perspectives. i) *How does an optimal approximation to softmax attention inspire linear attention design?* In this work, we mainly remove the Key matrix, use dynamic decay, and enhance local interactions and the token's own attention. This is clearly not the end of linear attention optimization. This work focuses on functional approximation, previous studies about the value approximation [30, 31, 50] can be further investigated on the basis of our functional approximation theory as well as MetaLA architecture. Additional optimization may include improving the recall ability of limited hidden states or designing better parameter functions. ii) *Does approximation to the softmax attention imply that linear attention has an upper capacity limit?* Taken literally, approximation seems to imply that linear attention cannot exceed softmax attention. However, we found better results for linear attention than softmax attention in some experimental results, such as zero-shot and LRA. Similar findings were also reported in previous work [13, 15, 16]. We argue that this issue deserves further exploration. For the time being, evaluation metrics that do not adequately reflect the model's capabilities, insufficient training [51], and linear attention that does have advantages in certain abilities [15] are all possibilities.

# 8   Acknowledgements

This work was partially supported by CAS Project for Young Scientists in Basic Research (YSBR-116), National Distinguished Young Scholars (62325603), National Natural Science Foundation of China (62236009, U22A20103, 62441606, 62406322), Beijing Science and Technology Plan (Z241100004224011), Beijing Natural Science Foundation for Distinguished Young Scholars (JQ21015), China Postdoctoral Science Foundation (GZB20240824, 2024M753497), and CAAI-MindSpore Open Fund, developed on OpenI Community.

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

## A1   Related Work

We overview several architectures of three types of linear models, i.e., LinFormer, SSM, and linRNN, and show how they can be specialized from the general form in Sec. 3.

### A1.1   LinFormer

LinFormer abandons the softmax form of vanilla attention, instead leveraging the dot product between Keys and Values to achieve linear complexity [9]. Advances in LinFormer mainly focus on designing better kernel function [30, 32, 52, 53, 54, 55, 50] such as the spiking neurons in a spiking neural network[56, 57, 58, 59] can be understood as efficient kernel functions[60, 61], exploring architectural optimization [13, 14, 62], introducing gating mechanism [15, 31, 63, 64], etc. We show the general form of several LinFormer architectures in Tab. A1. For the general form of LinFormers, $d_k = d_v = d$ is the most common choice.

• **Vanilla LinFormer** [9]: Vanilla linear Transformers lack explicit decay. Instead, They choose to maintain two hidden states containing a denominator to normalize attention scores. As for their general form, $\mathbf{q}_t/\mathbf{k}_t = f_{q/k}(\mathbf{x}_t, \theta_{q/k}) = \phi(\mathbf{x}_t \boldsymbol{W}_{Q/K})$ where $\theta_{q/k} = \boldsymbol{W}_{Q/K}$ and $\phi$ is the nonlinearity. $\mathbf{v}_t = \mathbf{x}_t \boldsymbol{W}_V$. There is no decay or output gate, i.e., $\mathbf{g}_t = \mathbf{1}$ and $\alpha_t = \mathbf{1}$.

• **RetNet** [14]: Different from vanilla LinFormer, RetNet uses positional encoding $\exp{(jn\theta)}$ and fixed decay $\lambda$ to control the hidden state. For its general form, $\mathbf{q}_t/\mathbf{k}_t/\mathbf{v}_t = f_{q/k/v}(\mathbf{x}_t, \theta_{q/k/v}) = \mathbf{x}_t \boldsymbol{W}_{Q/K/V}$ where $\theta_{q/k/v} = \boldsymbol{W}_{Q/K/V}$. $\alpha_t = \lambda \exp{(j\theta)}$ is the fixed decay vector, where $j$ is imaginary unit. Furthermore, $\mathbf{y}_t = [\mathbf{o}_t \odot \text{SiLU}(\mathbf{x}_t \boldsymbol{W}_G)]\boldsymbol{W}_O$ where $\theta_g = \boldsymbol{W}_G$.

• **TransNormer** [12]: TransNormer also uses positional encoding $\exp{(jn\theta)}$ and fixed decay $\lambda$ to control the hidden state. For its general form, $\mathbf{q}_t/\mathbf{k}_t = f_{q/k}(\mathbf{x}_t, \theta_{q/k}) = \phi(\mathbf{x}_t \boldsymbol{W}_{Q/K})$ where $\theta_{q/k} = \boldsymbol{W}_{Q/K}$ and $\phi$ is the nonlinearity. $\mathbf{v}_t = \mathbf{x}_t \boldsymbol{W}_V$, and $\alpha_t = \lambda \exp{(j\theta)}$ is the fixed decay vector, where $j$ is imaginary unit. The normalization layer chosen by the paper is SRMSNorm. Furthermore, $\mathbf{y}_t = [\mathbf{o}_t \odot (\mathbf{x}_t \boldsymbol{W}_U)]\boldsymbol{W}_O$ where $\theta_g = \boldsymbol{W}_U$.

• **GLA** [15]: GLA considers a data-dependent gating mechanism for LinFormer. For its general form, $\mathbf{q}_t/\mathbf{k}_t/\mathbf{v}_t = f_{q/k/v}(\mathbf{x}_t, \theta_{q/k/v}) = \mathbf{x}_t \boldsymbol{W}_{Q/K/V}$ where $\theta_{q/k/v} = \boldsymbol{W}_{Q/K/V}$. $\alpha_t = \text{sigmoid}(\mathbf{x}_t \boldsymbol{W}_\alpha^1 \boldsymbol{W}_\alpha^2 + \boldsymbol{b}_\alpha)^{1/\tau}$ is a dynamic decay, where $\boldsymbol{W}_\alpha^1$ and $\boldsymbol{W}_\alpha^2$ are low-rank matrices. Furthermore, $\mathbf{y}_t = [\mathbf{o}_t \odot \text{SiLU}(\mathbf{x}_t \boldsymbol{W}_r + \boldsymbol{b}_r)]\boldsymbol{W}_O$ where $\theta_g = [\boldsymbol{W}_r, \boldsymbol{b}_r]$.

Table A1: From general form to LinFormers.

| Models | Vanilla LinFormer[9] | RetNet[14] | TransNormer[12] | GLA[15] |
|---|---|---|---|---|
| $f_q(\mathbf{x}_t, \theta_q)$ | $\phi(\mathbf{x}_t \boldsymbol{W}_Q)$ | $\mathbf{x}_t \boldsymbol{W}_Q$ | $\phi(\mathbf{x}_t \boldsymbol{W}_Q)$ | $\mathbf{x}_t \boldsymbol{W}_Q$ |
| $f_k(\mathbf{x}_t, \theta_k)$ | $\phi(\mathbf{x}_t \boldsymbol{W}_K)$ | $\mathbf{x}_t \boldsymbol{W}_K$ | $\phi(\mathbf{x}_t \boldsymbol{W}_K)$ | $\mathbf{x}_t \boldsymbol{W}_K$ |
| $f_v(\mathbf{x}_t, \theta_v)$ | $\mathbf{x}_t \boldsymbol{W}_V$ | $\mathbf{x}_t \boldsymbol{W}_V$ | $\mathbf{x}_t \boldsymbol{W}_V$ | $\mathbf{x}_t \boldsymbol{W}_V$ |
| $f_\alpha(\mathbf{x}_t, \theta_\alpha)$ | $1$ | $\lambda \exp{(j\theta)}$ | $\lambda \exp{(j\theta)}$ | $\sigma(\mathbf{x}_t \boldsymbol{W}_\alpha^1 \boldsymbol{W}_\alpha^2 + \boldsymbol{b}_\alpha)$ |
| $f_g(\mathbf{x}_t, \theta_g)$ | $1$ | $\sigma(\mathbf{x}_t \boldsymbol{W}_G)$ | $\mathbf{x}_t \boldsymbol{W}_U$ | $\sigma(\mathbf{x}_t \boldsymbol{W}_r + \boldsymbol{b}_r)$ |
| Dimension | $d_k = d_v = d$ | | | $d_k = d/2$
$d_v = d$ |

### A1.2   SSM

SSM represents an alternative linear architecture to Transformer, based on the state space equations [24, 65]. Typical SSMs [25, 26, 66, 67, 68, 69] employ structured transition matrix and special initialization strategies such as HiPPO [27] to efficiently enhance long-range dependencies. Mamba [16] advances SSMs by introducing selection mechanism, i.e. input-dependent parameters [70] and designs a hardware-aware parallel algorithm. We show the general form of several SSM architectures in Table A2. All of them are SISO (Single-Input, Single-Output) except S5 [22]. We omit original S4 [10] model because it focuses on Diagonal Plus Low-Rank (DPLR) transition matrix

$\boldsymbol{A}$ rather than fully diagonal one, which is difficult to implement due to computational inefficiency. For most SSMs, $d_v = H$, which means each channel uses independent parameters $(\mathbf{q}_t^h, \alpha_t^h, \mathbf{k}_t^h)$.

• **DSS** [26]: DSS is the first research to show the effectiveness of diagonal structured SSM with fixed parameters, which means all parameters are constant through time. For the general form of DSS, $d_v = d = H, \frac{d_k}{H} = N$, where $N$ is an expansion. $\mathbf{Q}, \mathbf{K}, \boldsymbol{\Lambda}$ are independent of $\mathbf{x}_t$ and are determined by learnable parameters $[\boldsymbol{A}, \boldsymbol{B}, \boldsymbol{C}, \boldsymbol{\Delta}]$, using ZOH method to discretize. Furthermore, $\mathbf{x}_t = \mathbf{v}_t$, $\mathbf{g}_t = \mathbf{1}$ and $\boldsymbol{W}_O = \mathbf{I}$, where $\mathbf{I}$ denotes the identity matrix.

• **S4D** [25]: S4D is also a diagonal structured SSM with fixed parameters. It theoretically explain and expand the effectiveness of DSS and focus on parameterization and initialization of diagonal SSM. It can be discretized using ZOH or Bilinear method, resulting different $f_{k/\alpha}$ compared with DSS. Similarly, $\mathbf{x}_t = \mathbf{v}_t$, $\mathbf{g}_t = \mathbf{1}$ and $\boldsymbol{W}_O = \mathbf{I}$.

• **H3** [21]: H3 combines S4D with LinFormer and adds some architectural optimization such as local convolution and output gates. The core layer of H3 is S4D, and thus it shares the same general form with S4D.

• **S5** [22]: S5 uses MIMO (Multi-Input, Multi-Output) SSM with fixed parameters. For its general form, $d_v = Nd = H, \frac{d_k}{H} = 1$, where $N$ is an expansion much smaller than that of SISO models. $\alpha_t = \exp{(\Delta \boldsymbol{A})}$ is the fixed decay and $\mathbf{v}_t = f_v(\mathbf{x}_t, \theta_v) = \mathbf{x}_t \overline{\boldsymbol{B}}$ where $\overline{\boldsymbol{B}} = (\Delta \boldsymbol{A})^{-1}(\exp{(\Delta \boldsymbol{A})} - \mathbf{I}) \cdot \Delta \boldsymbol{B}$, i.e., it chooses ZOH method. Similar to previous SSM models, all the parameters are independent of $\mathbf{x}_t$ and are determined by learnable parameters $[\boldsymbol{A}, \boldsymbol{B}, \boldsymbol{C}, \boldsymbol{\Delta}]$. Furthermore, $\mathbf{q}_t = \mathbf{k}_t = \mathbf{g}_t = \mathbf{1}$ and $\boldsymbol{W}_O = \boldsymbol{C}$.

• **Mamba** [16]: Mamba advances SSMs with selection, which means all parameters are time-varying and input-dependent. For the general form of Mamba, $d_v = d = H, \frac{d_k}{H} = N$. $\mathbf{q}_t = \boldsymbol{C}_t = \mathbf{x}_t \boldsymbol{W}_C$. Similarly, $\mathbf{k}_t = \boldsymbol{B}_t = \mathbf{x}_t \boldsymbol{W}_B \Delta_t$. The decay term can be written as $\alpha_t = \exp(\Delta_t \boldsymbol{A})$. $\mathbf{x}_t = \mathbf{v}_t$, $\mathbf{g}_t = \mathbf{1}$ and $\boldsymbol{W}_O = \mathbf{I}$. Furthermore, $\Delta_t = \mathrm{softplus}(\mathbf{x}_t \boldsymbol{W}_{lora} + \boldsymbol{b}_\Delta)$.

Table A2: From general form to SSMs.

| Models | DSS[26], S4D[25], H3[21] (ZOH) | S4D[25], H3[21] (Bilinear) | Mamba[16] | S5[22] |
|---|---|---|---|---|
| $f_q(\mathbf{x}_t, \theta_q)$ | $\boldsymbol{C}$ | $\boldsymbol{C}$ | $\mathbf{x}_t \boldsymbol{W}_C$ | $\mathbf{1}$ |
| $f_k(\mathbf{x}_t, \theta_k)$ | $(\Delta \boldsymbol{A})^{-1}(\exp{(\Delta \boldsymbol{A})} - \mathbf{I})\Delta \boldsymbol{B}$ | $(\mathbf{I} - \frac{\Delta}{2}\boldsymbol{A})^{-1}\Delta \boldsymbol{B}$ | $\Delta_t(\mathbf{x}_t \boldsymbol{W}_B)$ | $\mathbf{1}$ |
| $f_v(\mathbf{x}_t, \theta_v)$ | $\mathbf{x}_t$ | $\mathbf{x}_t$ | $\mathbf{x}_t$ | $\mathbf{x}_t \overline{\boldsymbol{B}}$ |
| $f_\alpha(\mathbf{x}_t, \theta_\alpha)$ | $\exp{(\Delta \boldsymbol{A})}$ | $(\mathbf{I} - \frac{\Delta}{2}\boldsymbol{A})^{-1}(\mathbf{I} + \frac{\Delta}{2}\boldsymbol{A})$ | $\exp{(\Delta_t \boldsymbol{A})}$ | $\exp{(\Delta \boldsymbol{A})}$ |
| $f_g(\mathbf{x}_t, \theta_g)$ | $\mathbf{1}$ | $\mathbf{1}$ | $\mathbf{1}$ | $\mathbf{1}$ |
| Dimension | $d_v = d = H$ $d_k/H = N$ | | | $d_v = Nd = H$ $d_k/H = 1$ |

### A1.3 LinRNN

LinRNN is a variant of the vanilla Recurrent Neural Network (RNN) model [71, 72, 73] that eliminates the nonlinearity within the recurrence and employs element-wise product instead of matrix multiplication [11, 28, 74, 75]. Recent works have made efforts to explore more effective recurrence [17, 19, 76, 77] and design more advanced gating mechanisms [18, 20, 33]. We show the general form of several LinRNN architectures in Tab. A3. LinRNNs generally implement $d_k = d_v = H = d$ except LRU [19], which means each channel uses independent parameters $(\mathbf{q}_t^h, \alpha_t^h, \mathbf{k}_t^h)$. They maintain a smaller unexpanded hidden state, which is a 1D vector and can be seen as a special case with head dimension size $d_k' = 1$.

• **RWKV-4** [17]: RWKV-4 is a linear RNN model which includes fixed decay and output gates. Moreover, RWKV-4 treats the present token differently and uses token shift operation (we omit for clear illustration). For its general form, $\mathbf{k}_t = \exp{(\mathbf{x}_t \boldsymbol{W}_K)}$ where $\theta_k = \boldsymbol{W}_K$. $\mathbf{v}_t = \mathbf{x}_t \boldsymbol{W}_V$ while $\mathbf{q}_t = \mathbf{1}$. $\alpha_t = \exp{(-\boldsymbol{W})}$ is fixed decay with the learnable parameter $\theta_\alpha = \boldsymbol{W}$. Furthermore, $\mathbf{y}_t = [\mathbf{o}_t \odot \mathrm{sigmoid}(\mathbf{x}_t \boldsymbol{W}_r)]\boldsymbol{W}_O$ where $\theta_g = \boldsymbol{W}_r$.

- **GLRU** [20]: Gated linear recurrent unit (GLRU) mentioned in HGRN is a linear RNN model including input/forget/output gates. For the general form of GLRU, $\alpha_t/\mathbf{k}_t = \mathrm{f}_{\alpha/k}(\mathbf{x}_t, \theta_{\alpha/k}) = \sigma(\mathbf{x}_t \boldsymbol{W}_{\alpha/K} + \boldsymbol{b}_{\alpha/K})$ where $\theta_{\alpha/k} = [\boldsymbol{W}_{\alpha/K}, \boldsymbol{b}_{\alpha/K}]$ and $\sigma$ is the sigmoid function. $\mathbf{v}_t = \mathrm{SiLU}(\mathbf{x}_t \boldsymbol{W}_V + \boldsymbol{b}_V)$ while $\mathbf{q}_t = \mathbf{1}$. Furthermore, $\mathbf{y}_t = [\mathbf{o}_t \odot \mathrm{SiLU}(\mathbf{x}_t \boldsymbol{W}_g + \boldsymbol{b}_g)]\boldsymbol{W}_O$ where $\theta_g = [\boldsymbol{W}_g, \boldsymbol{b}_g]$.

- **Griffin** [18]: Griffin is an RNN model with gated linear recurrence. Moreover, it ties dynamic decay and Key as we do. For the general form of Griffin, $\alpha_t = \mathrm{f}_\alpha(\mathbf{x}_t, \theta_\alpha) = \boldsymbol{A}^{c\cdot\sigma(\mathbf{x}_t \boldsymbol{W}_a + \boldsymbol{b}_a)}$ where $\theta_\alpha = [\boldsymbol{A}, \boldsymbol{W}_a, \boldsymbol{b}_a]$. Then $\mathbf{k}_t = \sqrt{1 - \alpha_t^2}$ to replace Key's role. $\mathbf{v}_t = \sigma(\mathbf{x}_t \boldsymbol{W}_x + \boldsymbol{b}_x) \odot \mathbf{x}_t$ which contains an input gate additionally. $\mathbf{q}_t = \mathbf{1}$ and $\mathbf{y}_t = [\mathbf{o}_t \odot \mathrm{GeLU}(\mathbf{x}_t \boldsymbol{W}_g + \boldsymbol{b}_g)]\boldsymbol{W}_O$ where $\theta_g = [\boldsymbol{W}_g, \boldsymbol{b}_g]$.

- **LRU** [19]: LRU is a linear and diagonal RNN model with fixed decay, inspired by deep SSMs' success. For its general form, $d_v = Nd = H, \frac{d_k}{H} = 1$, where $N$ is a small expansion. $\alpha_t = \boldsymbol{A}$ is the fixed decay and $\mathbf{v}_t = \mathrm{f}_v(\mathbf{x}_t, \theta_v) = \mathbf{x}_t \boldsymbol{B}$ where $\theta_v = \boldsymbol{B}$. Furthermore, $\mathbf{q}_t = \mathbf{k}_t = \mathbf{g}_t = \mathbf{1}$ and $\boldsymbol{W}_O = \boldsymbol{C}$.

Table A3: From general form to LinRNN.

| Models | RWKV-4[17] | GLRU[20] | Griffin[18] | LRU[19] |
|---|---|---|---|---|
| $\mathrm{f}_q(\mathbf{x}_t, \theta_q)$ | $1$ | $1$ | $1$ | $1$ |
| $\mathrm{f}_k(\mathbf{x}_t, \theta_k)$ | $\exp(\mathbf{x}_t \boldsymbol{W}_K)$ | $\sigma(\mathbf{x}_t \boldsymbol{W}_i + \boldsymbol{b}_i)$ | $\sqrt{1 - \alpha_t^2}$ | $1$ |
| $\mathrm{f}_v(\mathbf{x}_t, \theta_v)$ | $\mathbf{x}_t \boldsymbol{W}_V$ | $\phi(\mathbf{x}_t \boldsymbol{W}_c + \boldsymbol{b}_c)$ | $\sigma(\mathbf{x}_t \boldsymbol{W}_x + \boldsymbol{b}_x) \odot \mathbf{x}_t$ | $\mathbf{x}_t \boldsymbol{B}$ |
| $\mathrm{f}_\alpha(\mathbf{x}_t, \theta_\alpha)$ | $\exp(-\boldsymbol{W})$ | $\sigma(\mathbf{x}_t \boldsymbol{W}_f + \boldsymbol{b}_f)$ | $\boldsymbol{A}^{c\cdot\sigma(\mathbf{x}_t \boldsymbol{W}_a + \boldsymbol{b}_a)}$ | $\boldsymbol{A}$ |
| $\mathrm{f}_g(\mathbf{x}_t, \theta_g)$ | $\sigma(\mathbf{x}_t \boldsymbol{W}_r)$ | $\phi(\mathbf{x}_t \boldsymbol{W}_g + \boldsymbol{b}_g)$ | $\sigma(\mathbf{x}_t \boldsymbol{W}_g + \boldsymbol{b}_g)$ | $1$ |
| Dimension | | $d_k = d_v = d = H$ | | $d_v = Nd = H$ 
 $d_k/H = 1$ |

### A1.4 Approximation to Softmax Attention

Approximation to softmax attention includes two parts: value and functional approximation. Some previous works of LinFormer [31, 30, 53, 78] were devoted to designing better kernel map $\phi$ to approximate the value of SoftAttMap via randomized features, i.e., $\phi(\mathbf{q}_t)\phi^\top(\mathbf{k}_s) \approx \exp(\mathbf{q}_t \mathbf{k}_s^\top)$. Recent work [50] further introduces attention weight distillation loss to minimize the cross-entropy loss between the computed linear attention weights and those that would have been computed via softmax attention, in order to achieve strict value approximation. Most of them follow typical designs in [9] and use only $\mathbf{Q}$ and $\mathbf{K}$. However, in this work, we find that only considering value approximation is insufficient. Different from previous studies, we investigate functional approximation of SoftAttMap. Through theoretical analysis, we show that these works cannot achieve the functionality of softmax attention due to lack of dynamic decay $\boldsymbol{\Lambda}_t$. Therefore, their purpose of achieving value approximation cannot be achieved as well. It is worth noting that this is not a denial of the significance of previous works about value approximation. We can also get inspiration from them. Functional approximation is the basis and prerequisite for value approximation.

## A2 Analysis of Optimal Linear Approximation Conditions

In this section, we show that existing linear models are not optimal linear approximation to softmax attention map, according to definition 4.1, and prove that linear models with only Query and dynamic decay can satisfy the optimal linear approximation conditions. In definition 4.1, C0 underlines computational and memory efficiency. C1 and C2 consider memory and modeling ability of linear attention. C3 is our expectation to seek the least parameters on the premise that previous three conditions are met. We first discuss the requirements to satisfy each conditions, based on the general form of linear attention in Sec. 3. All the linear models own linear complexity and satisfy C0, so we analyze C1, C2 and C3 in appendix A2.1, appendix A2.2 and appendix A2.3, respectively. Then we synthesize all the discussions and present the conclusions in appendix A2.4.

For notation we use $\mathbf{Q}, \mathbf{K}, \mathbf{V}, \mathbf{\Lambda}_t$ to denote Query, Key, Value and Decay matrices respectively. Considering decay can be either dynamic or fixed, here we use subscript $t$ to distinguish, i.e., $\mathbf{\Lambda}/\mathbf{\Lambda}_t$ denote fixed/dynamic decay. Corresponding $\mathbf{q}_t, \mathbf{k}_t, \mathbf{v}_t, \alpha/\alpha_t$ are query, key, value and fixed/dynamic decay of each timestep.

## A2.1 Analysis of Dynamic Memory Ability (C1)

**Proposition A2.1.** *Only models with dynamic decay can satisfy C1 (Dynamic memory ability). Let* $\mathbf{S}_t \in \mathcal{R}^{d_k \times d_v}$ *be hidden state of general linear attention (see Sec. 3). At a time $t$, the information about* $\mathbf{v}_{t_1}, \ldots, \mathbf{v}_{t_{d_k}}$ *is successfully stored in* $\mathbf{S}_t$ $(t_1, \ldots, t_{d_k} \leq t)$, *which means* $(\mathbf{S}_t)_{i,:} = \mathbf{v}_{t_i}$. *When the new important input* $\mathbf{v}_{t+1}$ *arrives, only models with dynamic decay can substitute historical unimportant* $\mathbf{v}_{t_1}$ *by* $\mathbf{v}_{t+1}$ *successfully, i.e., obtain* $(\mathbf{S}_{t+1})_{1,:} = \mathbf{v}_{t+1}$.

*Proof.* Without loss of generality, suppose $t = d_k$ and $t_i = i (i = 1, \ldots, d_k)$, where $d_k$ is Key dimension. That is, the information about $\mathbf{v}_1, \ldots, \mathbf{v}_{d_k}$ is successfully stored, which means the $i$-th row of $\mathbf{S}_{d_k}$, $(\mathbf{S}_{d_k})_{i,:} = \mathbf{v}_i$. Suppose $\mathbf{v}_1$ is unimportant information need to be substituted. Now the new and important input $\mathbf{v}_{d_k+1}$ arrives, models with general form Eq. (13) update $\mathbf{S}_{d_k}$ as follows (heads are omitted for simplicity):

$$\mathbf{S}_{d_k+1} = \mathrm{diag}(\alpha_{d_k+1})\mathbf{S}_{d_k} + (\mathbf{k}_{d_k+1})^\top \mathbf{v}_{d_k+1}. \tag{A1}$$

We will check whether the model can obtain $(\mathbf{S}_{d_k+1})_{1,:} = \mathbf{v}_{d_k+1}$, with fixed/dynamic decay or without decay. In the following discussion we let $\mathbf{k}_{d_k+1} = \mathbf{1}$.

i) *Models with fixed decay*, which means $\alpha_{d_k+1} = \alpha \in (0, 1)$. The model can update and obtain $(\mathbf{S}_{d_k+1})_{1,:} = \alpha \mathbf{v}_1 + \mathbf{v}_{d_k+1}$. This means the model with fixed decay can only store most recent several tokens rather than store most important tokens without information loss, i.e., it has ability to forget but no ability to forget and memorize dynamically.

ii) *Models without decay*, which means $\alpha_{d_k+1} = 1$. This case results in $(\mathbf{S}_{d_k+1})_{1,:} = \mathbf{v}_1 + \mathbf{v}_{d_k+1}$, which means they have no mechanism to erase old information. So such linear models have no ability to forget and memorize dynamically, and what they can only do is information blending. Thus the attention at time $t$ is distracted by all the information before, making it hard to focus on and approximate most important ones. After normalization, the attention distribution will have a relative high entropy along time dimension, which is often referred as attention dilution problem.

iii) *Models with dynamic decay*, which means $\alpha_{d_k+1} \in [0, 1]$. The model can easily erase and substitute information. Setting $\alpha_{d_k+1} = [0, 1, \cdots, 1]$ can obtain $(\mathbf{S}_{d_k+1})_{1,:} = \mathbf{v}_{d_k+1}$. So such linear models have ability to memorize and forget dynamically, making historical unimportant information have few effect on new one. □

## A2.2 Analysis of Static Approximation Ability (C2)

Eq. (18) illustrates the LinAttMap only relate to $\mathbf{q}_t = \mathrm{f}_q(\mathbf{x}_t, \theta_q), \mathbf{k}_t = \mathrm{f}_k(\mathbf{x}_t, \theta_k)$ and $\alpha_t = \mathrm{f}_\alpha(\mathbf{x}_t, \theta_\alpha)$. Assuming the inputs are good enough and the functions $(\mathrm{f}_q, \mathrm{f}_k, \mathrm{f}_\alpha)$ are expressive enough, we can shift from solving $(\theta_q, \theta_k, \theta_\alpha)$ to solving $(\mathbf{Q}, \mathbf{K}, \mathbf{\Lambda}_t)$. Based on definition 4.1, we focus on approximating the attention scores between stored tokens $\mathbf{x}_{t_1}, \ldots, \mathbf{x}_{t_{d_k}}$.

**Proposition A2.2.** *Only models with parameters* $(\mathbf{Q}, \mathbf{K}, \mathbf{\Lambda}_t)$, $(\mathbf{Q}, \mathbf{K})$ *or* $(\mathbf{Q}, \mathbf{\Lambda}_t)$ *can satisfy C2 (Static approximation ability). For an arbitrarily given softmax attention map* $\mathbf{P}$ *with scores* $p_{ts}$, *only above models can ensure Eq. (A2) and Eq. (A3) hold with bounded parameters.*

$$f(\mathbf{x}_t, \mathbf{x}_s | \mathbf{Q}, \mathbf{K}, \mathbf{\Lambda}_t) = \mathbf{q}_t \cdot \left( \left( \prod_{j=s+1}^{t} \alpha_j \right) \odot \mathbf{k}_s \right)^\top = p_{ts}, \forall s \leq t = t_1, \cdots, t_{d_k}, \tag{A2}$$

$$\mathrm{s.t.} \quad ||\mathbf{q}_t|| \leq C_q, ||\mathbf{k}_t|| \leq C_k, ||\alpha_t|| \in [0, 1], \forall t = t_1, \cdots, t_{d_k}, \tag{A3}$$

*where* $C_q, C_k$ *are constant and* $|| \cdot ||$ *denotes vector norm. In our general form in Sec. 3,* $f(\mathbf{x}_t, \mathbf{x}_s | \mathbf{Q}, \mathbf{K}, \mathbf{\Lambda}_t) = \mathrm{LinAttMap}(\mathbf{Q}, \mathbf{K})_{t,s}$ *(see Eq. (18)). For bounded inputs* $\mathbf{X}$, *bounded parameters* $(\theta_q, \theta_k, \theta_\alpha)$ *are equivalent to bounded* $(\mathbf{Q}, \mathbf{K}, \mathbf{\Lambda}_t)$.

*Proof.* Without loss of generality, suppose $t = d_k$ and $t_i = i (i = 1, \ldots, d_k)$, where $d_k$ is Key dimension. That is, we need to prove following equations hold (heads are omitted for simplicity):

$$f(\mathbf{x}_t, \mathbf{x}_s | \mathbf{Q}, \mathbf{K}, \boldsymbol{\Lambda}_t) = \mathbf{q}_t \cdot \left( \left( \prod_{j=s+1}^{t} \alpha_j \right) \odot \mathbf{k}_s \right)^{\top} = p_{ts}, \forall s \le t = 1, \cdots, d_k, \tag{A4}$$

$$\text{s.\,t.} \quad ||\mathbf{q}_t|| \le C_q, ||\mathbf{k}_t|| \le C_k, ||\alpha_t|| \in [0, 1], \forall t = 1, \cdots, d_k, \tag{A5}$$

**Simple case.** We first simplify the problem via i) setting dimension size of $\mathbf{Q}$, i.e. $d_k = 1$ and ii) considering only a given time $t$ $(1 \le t \le d_k)$ and its autoregressive attention distribution $\mathbf{p}_t = [p_{ts}, s = 1, \ldots, t] \in \mathcal{R}^{1 \times t}$. We need to prove the following equations hold with bounded parameters, as a foundation conclusion:

$$f(\mathbf{x}_t, \mathbf{x}_s | \mathbf{Q}, \mathbf{K}, \boldsymbol{\Lambda}_t) = q_t \left( \prod_{j=s+1}^{t} \alpha_j \right) k_s = p_{ts}, \forall s = 1, \ldots, t, \tag{A6}$$

$$\text{s.\,t.} \quad |q_s| \le C_q, |k_s| \le C_k, \alpha_s \in [0, 1], \forall s = 1, \ldots, t. \tag{A7}$$

We discuss models with fixed/dynamic decay or without decay. For cases with dynamic decay, we further consider using dynamic decay to replace key. Query's role is discussed later.

i) *Models with fixed decay $\alpha$ result in unbounded parameters*. We can solve Eq. (A6) and derive:

$$|k_s| = \frac{p_{ts}}{\alpha^{t-s}|q_t|} \ge \frac{p_{ts}}{C_q} \cdot \frac{1}{\alpha^{t-s}}, \forall s = 1, \ldots, t. \tag{A8}$$

Because $\alpha \in (0, 1)$, the last term $\frac{1}{\alpha^{t-s}}$ is unbounded, which leads to unbounded $k_s$. Hence Eq. (A7) can not be satisfied. The result is intuitive because the fixed exponential decay makes it hard for a token to attend to distant information, leading to an excessively focused attention map rather than an arbitrary attention map. This conclusion about fixed decay is general and is unrelated to the usage of $\mathbf{Q}$ or $\mathbf{K}$.

ii) *Models without decay ($\alpha = 1$) can satisfy Eq. (A6)*. We can solve Eq. (A6) and derive:

$$k_s = \frac{p_{ts}}{q_t}, \forall s = 1, \ldots, t. \tag{A9}$$

At any time $s$, fix an appropriate $|q_t| \le C_q$, and we can obtain bounded solutions for all $k_s$.

iii) *Models with both dynamic decay $\alpha_t$ and key $k_t$ can satisfy Eq. (A6), but have parameter-redundancy*. We can solve Eq. (A6) starting from time $t$ to time 1:

$$k_t = \frac{p_{tt}}{q_t}, \tag{A10}$$

$$k_{t-1}\alpha_t = \frac{p_{t\,t-1}}{q_t}, \tag{A11}$$

$$\cdots, \tag{A12}$$

$$k_{t-k}\alpha_{t-k+1} \cdot (\alpha_{t-k+2} \cdots \alpha_t) = \frac{p_{t\,t-k}}{q_t}, \tag{A13}$$

$$\cdots, \tag{A14}$$

$$k_1\alpha_2 \cdot (\alpha_3, \cdots, \alpha_t) = \frac{p_{t1}}{q_t}. \tag{A15}$$

At any time $t - k$, parameters $\alpha_{t-k+2}, \cdots, \alpha_t$ have already been determined. It is equivalent to use two free parameters $k_{t-k}, \alpha_{t-k+1}$ to concurrently approximate just one variable $\frac{p_{t\,t-k}}{q_t}$, which is redundant. Dynamic decay and key actually have similar function, i.e., to balance historical memory and new input. Fix a $|q_t| \le C_q$, we can obtain many possible bounded solutions of all $k_s$ and $\alpha_s$.

iv) *Models with dynamic decay $\alpha_t$ and without key $k_t$ can satisfy Eq. (A6)*. Inspired by the parameter-redundancy in iii), we further consider a linear model with only query and dynamic decay, i.e., using dynamic decay $1 - \alpha_t$ to replace $k_t$'s role to approximate softmax attention map. We further assume $q_t = 1$, which is bounded, and define $S_t = \sum_{s=1}^{t} p_{ts}$, which satisfies $S_t \in [0, 1]$ and $S_t \ge S_{t-1}$.

Solve Eq. (A6) starting from time $t$ to time 1:

$$\alpha_t = 1 - p_{tt} = S_{t-1}, \tag{A16}$$

$$\alpha_{t-1} = 1 - \frac{p_{t\,t-1}}{\alpha_t} = \frac{S_{t-2}}{S_{t-1}}, \tag{A17}$$

$$\cdots, \tag{A18}$$

$$\alpha_{t-k} = 1 - \frac{p_{t\,t-k}}{\alpha_{t-k+1}\cdots\alpha_t} = \frac{S_{t-k-1}}{S_{t-k}}, \tag{A19}$$

$$\cdots, \tag{A20}$$

$$\alpha_1 = 1 - \frac{p_{t1}}{\alpha_2\cdots\alpha_t} = \frac{S_0}{S_1}. \tag{A21}$$

At any time $t-k$, we can derive a closed-form and bounded solution $\alpha_{t-k} = \frac{S_{t-k-1}}{S_{t-k}} \in [0,1]$.

**General case.** Aiming to solve Eq. (A4) and Eq. (A5), we generalize to vector version with $d_k > 1$ and consider distribution of all time $\mathbf{p}_t, t = 1, \ldots, d_k$. In an extreme case, let one dimension of $\mathbf{q}$ relate to one time, which means $\mathbf{q}_t = [0, \cdots, q_t, \cdots, 0]$. Through the analysis above for one dimension (Eq. (A6) and Eq. (A7)), we can ensure:

$$\mathbf{q}_t \cdot \left( \left( \prod_{j=s+1}^{t} \alpha_j \right) \odot \mathbf{k}_s \right)^{\top} = q_t \left( \prod_{j=s+1}^{t} \alpha_j \right) k_s = p_{ts}, \forall s = 1, \ldots, t. \tag{A22}$$

Actually, we can further relax the condition and view $\mathbf{q}_t$ as a selector, which selects several channels of Hadamard product of $\alpha_j$ and $\mathbf{k}_s$, and uses these channels to approximate its attention distribution. One-hot assumption of $\mathbf{q}_t$ is an extremely hard selector. This means models without $\mathbf{Q}$ can not approximate softmax attention map, because without selection they can only fit one token's attention distribution theoretically.

In summary, through analysis of simple case i) to iv), models with fixed decay $\mathbf{\Lambda}$ cannot ensure bounded parameters. Through analysis of general case, $\mathbf{Q}$ is necessary. So we can conclude that: only models with parameters $(\mathbf{Q}, \mathbf{K}, \mathbf{\Lambda}_t)$, $(\mathbf{Q}, \mathbf{K})$ or $(\mathbf{Q}, \mathbf{\Lambda}_t)$ can satisfy C2. (For models with only $\mathbf{Q}$, without decay and $\mathbf{k}_t = \mathbf{1}$ for all $t$, a token's attentions to other tokens are all the same, which means such models can only approximate one special attention map.)

$\square$

### A2.3 Analysis of Least Parameter Approximation (C3)

When model dimension $d, d_k, d_v$ are fixed, parameter counts of $\theta_q, \theta_k, \theta_\alpha$ are fixed too. Thus fewest parameters means fewest parameter groups. Linear attentions with general form can be classified into three types based on the parameter groups: i) using $(\mathbf{Q}, \mathbf{K}, \mathbf{\Lambda}/\mathbf{\Lambda}_t)$ all together, ii) exploiting $(\mathbf{Q}, \mathbf{K})$, $(\mathbf{Q}, \mathbf{\Lambda}/\mathbf{\Lambda}_t)$ or $(\mathbf{K}, \mathbf{\Lambda}/\mathbf{\Lambda}_t)$, iii) employing only one of $\mathbf{Q}, \mathbf{K}, \mathbf{\Lambda}/\mathbf{\Lambda}_t$.

**Proposition A2.3.** *Only models with Query and dynamic decay* $(\mathbf{Q}, \mathbf{\Lambda}_t)$ *can satisfy C3 (Least parameter approximation). Models with* $(\mathbf{Q}, \mathbf{\Lambda}_t)$ *can meet C0, C1, C2 simultaneously with fewest parameters.*

*Proof.* i) C0: all the linear models with general form (Sec. 3) own linear complexity and satisfy C0.

ii) C1: according to proposition A2.1, only models with dynamic decay $\mathbf{\Lambda}_t$ have dynamic memory ability.

iii) C2: according to proposition A2.2, only models with parameters $(\mathbf{Q}, \mathbf{K}, \mathbf{\Lambda}_t)$, $(\mathbf{Q}, \mathbf{K})$ or $(\mathbf{Q}, \mathbf{\Lambda}_t)$ have static approximation ability.

So we claim that only models using $(\mathbf{Q}, \mathbf{K}, \mathbf{\Lambda}_t)$ or $(\mathbf{Q}, \mathbf{\Lambda}_t)$ can meet C0, C1 and C2 simultaneously. And $(\mathbf{Q}, \mathbf{\Lambda}_t)$ with two parameter groups is least parameter approximation. That is, dynamic decay can replace Key's function to balance historical information and new input when approximation is performed. $\square$

### A2.4 Conclusions of Optimal Linear Approximation Analysis

In this section, we summarize our conclusions of theoretical analysis. It is worth noting that our theory studies functions of LinAttMap, but not whether these functions will be successfully learned.

i) *Linear approximation.* The necessary conditions (C1 and C2) for LinAttMap to achieve approximation to SoftAttMap is that its implementation must include at least two parameter groups: Query $\mathbf{Q}$ and dynamic decay $\boldsymbol{\Lambda}_t$. Both $(\mathbf{Q}, \mathbf{K}, \boldsymbol{\Lambda}_t)$ and $(\mathbf{Q}, \boldsymbol{\Lambda}_t)$ can achieve approximation.

ii) *Least parameter approximation.* LinAttMap utilizing $(\mathbf{Q}, \boldsymbol{\Lambda}_t)$ can achieve linear approximation to SoftAttMap with theoretically fewest parameters.

iii) *Function of Query.* $\mathbf{Q}$ can be seen as a channel selector, which selects several channels of Hadamard product of $\alpha_t$ and $\mathbf{k}_t$, and uses these channels to approximate attention map. Thus it is indispensable for approximation and its dimension size (also the model memory size) is the guarantee of approximation ability.

iv) *Function of dynamic decay.* Dynamic decay $\boldsymbol{\Lambda}_t$ is the key to achieve dynamic memory and precise approximation, and can substitute the role of $\mathbf{K}$ when approximation is performed.

## A3 MetaLA Architecture

We here introduce Meta Linear Attention (MetaLA), a linear approximation with least parameters to softmax attention map. We design three enhancements of MetaLA relative to the general linear attention in Sec. 3: i) The Key matrix is not used, which is based on our theoretical analysis. ii) Self-augmentation and iii) Short convolution are two other optional techniques to further enhance approximation ability of our model. appendix A3.1 explicates enhancement i) and design of basic MetaLA layer. appendix A3.2 introduces enhancements ii) and iii), and design of overall MetaLA architecture.

### A3.1 MetaLA Layer

In MetaLA layer, we exploit $1 - \alpha_t$ to replace $\mathbf{k}_t$ based on theoretical analysis in appendix A2, i.e., LinAttMap utilizing $(\mathbf{Q}, \boldsymbol{\Lambda}_t)$ is the linear approximation with least parameter redundancy to SoftAttMap and $\mathbf{K}$ is redundant.

**General Recurrent Form.** Using marks in Eq. (11)-Eq. (15), the improvement lies in Eq. (A25):

$$\mathbf{q}_t = \mathbf{x}_t \boldsymbol{W}_Q, \alpha_t = \sigma(\mathbf{x}_t \boldsymbol{W}_\alpha) \quad \in \mathcal{R}^{1 \times d_k}, \tag{A23}$$

$$\mathbf{v}_t = \mathbf{x}_t \boldsymbol{W}_V, \mathbf{g}_t = \phi(\mathbf{x}_t \boldsymbol{W}_G + \boldsymbol{b}_G) \quad \in \mathcal{R}^{1 \times d_v}, \tag{A24}$$

$$\mathbf{S}_t^h = \mathrm{diag}(\alpha_t^h) \mathbf{S}_{t-1}^h + (\mathbf{1} - \alpha_t^h)^\mathsf{T} \mathbf{v}_t \quad \in \mathcal{R}^{d_k' \times d_v'}, \tag{A25}$$

$$\mathbf{o}_t = \mathrm{XNorm}(\mathrm{concat}[\mathbf{q}_t^1 \mathbf{S}_t^1, \mathbf{q}_t^2 \mathbf{S}_t^2, \cdots, \mathbf{q}_t^H \mathbf{S}_t^H]) \quad \in \mathcal{R}^{1 \times d_v}, \tag{A26}$$

$$\mathbf{y}_t = (\mathbf{g}_t \odot \mathbf{o}_t) \boldsymbol{W}_O \quad \in \mathcal{R}^{1 \times d}, \tag{A27}$$

where $\mathbf{x}_t \in \mathcal{R}^{1 \times d}$ denotes the current input. $\boldsymbol{W}_Q, \boldsymbol{W}_\alpha \in \mathcal{R}^{d \times d_k}$, $\boldsymbol{W}_V, \boldsymbol{W}_G \in \mathcal{R}^{d \times d_v}$ and $\boldsymbol{W}_O \in \mathcal{R}^{d_v \times d}$ are learnable parameters. $d_{k/v}' = \frac{d_{k/v}}{H}$ and $h = 1, \cdots, H$ is the index of heads. Here $d_k$ represents Query/Decay dimension. $\mathrm{LayerNorm}$ is chosen as the normalization operation. Furthermore, we choose $\sigma = \mathrm{Sigmoid}(\cdot)^{1/\tau}$ following [15], where $\tau = 16$ is used to control the value of dynamic decay and $\phi = \mathrm{SiLU}$.

As for parameter allocation, we simply set $d_v = d$ and $d_k = \frac{d}{2}$, similar to GLA [15]. Thus, the whole number of parameters used by a MetaLA layer is $4d^2$, the same as the softmax attention layer and smaller than other popular attention-like subquadratic models such as RetNet [14] ($8d^2$) and GLA [15] ($4d^2 + 24d$), etc. Without usage of $\mathbf{K}$, we can allocate more parameters and utilize full-rank matrix $\boldsymbol{W}_\alpha$ to produce dynamic decay rather than low-rank matrix used by GLA, such that do not sacrifice expression capacity of $\mathrm{f}_\alpha$ and approximation performance of MetaLA.

**General Parallel Form.** The recurrent form of MetaLA can be written in a parallel mode as follows:

$$\mathbf{O} = \Big(\Big((\mathbf{Q} \odot \mathbf{A})\big(\tfrac{\mathbf{B}}{\mathbf{A}}\big)^{\top}\Big) \odot \mathbf{M}\Big)\mathbf{V}, \tag{A28}$$

$$(\mathbf{Q}/\mathbf{K}/\mathbf{V})_{t,:} = (\mathbf{q}/\mathbf{k}/\mathbf{v})_t, \quad \mathbf{A}_{t,:} = \prod_{j=1}^{t} \alpha_j, \quad \mathbf{B}_{t,:} = \mathbf{1} - \alpha_t, \quad \mathbf{M}_{i,j} = \begin{cases} 1, i \le j. \\ 0, i > j. \end{cases} \tag{A29}$$

$\tfrac{\mathbf{B}}{\mathbf{A}}$ in Eq. (A28) denotes element-wise division and $(\mathbf{Q})_{t,:}$ means the $t$-th row of matrix $\mathbf{Q}$. In practical implementation, we utilize chunkwise form proposed in [15] for hardware-efficient training.

**Attention map.** Here the attention map is $\Big(\big((\mathbf{Q} \odot \mathbf{A})\big(\tfrac{\mathbf{B}}{\mathbf{A}}\big)^{\top}\big) \odot \mathbf{M}\Big)$. The element in the attention map matrix is as follows (heads are omitted for simplicity):

$$\mathrm{LinAttMap}_{t,s} = \begin{cases} \mathbf{q}_t \cdot \Big(\big(\prod_{j=s+1}^{t} \alpha_j\big) \odot (\mathbf{1} - \alpha_s)^{\top}\Big), s \le t. \\ 0, s > t. \end{cases} \tag{A30}$$

### A3.2 MetaLA Transformer

**Improved MetaLA.** After deriving the basic MetaLA layer shown above, we consider further optimization when designing complete MetaLA block. One observation is that in most cases, setting $\alpha_t = 0$ and completely discarding historical information is catastrophic. Actually, in real-world scenario, the learned dynamic decay $\alpha_t$ always closes to 1 because of many tasks' strong need to capture long-term dependencies. Hence we propose a technique called **self-augmentation** to enlarge a token's attention to itself while do not disrupt the model's state constitution, in order to better avert attention dilution. Moreover, an additional **short convolution** can be inserted before entering MetaLA layer to further enhance local interaction, motivated by Mamba [16] and Griffin [18]. Combining these two techniques, the improved MetaLA is shown as follows, the main improvements lie in Eq. (A31) and Eq. (A35):

$$\mathbf{x}_t = \mathrm{Conv1d}(\mathbf{x}_t) \quad \in \mathcal{R}^{1 \times d}, \tag{A31}$$

$$\mathbf{q}_t = \mathbf{x}_t \boldsymbol{W}_Q, \alpha_t = \sigma(\mathbf{x}_t \boldsymbol{W}_\alpha) \quad \in \mathcal{R}^{1 \times d_k}, \tag{A32}$$

$$\mathbf{v}_t = \mathbf{x}_t \boldsymbol{W}_V, \mathbf{g}_t = \phi(\mathbf{x}_t \boldsymbol{W}_G) \quad \in \mathcal{R}^{1 \times d_v}, \tag{A33}$$

$$\mathbf{S}_t^h = \mathrm{diag}(\alpha_t^h)\mathbf{S}_{t-1}^h + (1 - \alpha_t^h)^{\mathsf{T}}\mathbf{v}_t \quad \in \mathcal{R}^{d_k' \times d_v'}, \tag{A34}$$

$$\mathbf{o}_t^h = \mathbf{q}_t^h \mathbf{S}_t^h + \sigma_{\mathrm{aug}}\big(\mathbf{q}_t^h (\boldsymbol{w}_{\mathrm{aug}}^h \odot (1 - \alpha_t^h))^{\mathsf{T}} \mathbf{v}_t\big) \quad \in \mathcal{R}^{1 \times d_v'}, \tag{A35}$$

$$\mathbf{o}_t = \mathrm{XNorm}(\mathrm{concat}[\mathbf{o}_t^1, \mathbf{o}_t^2, \cdots, \mathbf{o}_t^H]) \quad \in \mathcal{R}^{1 \times d_v}, \tag{A36}$$

$$\mathbf{y}_t = (\mathbf{g}_t \odot \mathbf{o}_t)\boldsymbol{W}_O \quad \in \mathcal{R}^{1 \times d}. \tag{A37}$$

As for self-augmentation, without changing the composition of hidden state $\mathbf{S}_t^h$, it is only added on the output process through a learnable parameter $\boldsymbol{w}_{\mathrm{aug}} \in \mathcal{R}^{1 \times d_k}$, which is then divided into heads like other parameters do. $\sigma_{\mathrm{aug}}(\cdot)$ is a nonlinearity to control the magnitude of augmentation term and avoid covering with the original attention. We choose $\mathrm{Sigmoid}(\cdot)$ in this paper.

The proposed design has two advantages: First, it maintains parallel computing as shown in Eq. (A38); Second, it only enhances current token's own attention and does not change the attention of future tokens to the current token ($p_{ts}, s < t$), because we only change $\mathbf{o}_t^h$ while maintaining $\mathbf{S}_t^h$ unchanged like that in original MetaLA layer Eq. (A25). Thus the separation of output and memory is realized.

$$\mathbf{O} = \Big(\Big((\mathbf{Q} \odot \mathbf{A})\big(\tfrac{\mathbf{B}}{\mathbf{A}}\big)^{\top}\Big) \odot \mathbf{M}\Big)\mathbf{V} + \mathrm{diag}\big(\mathrm{sum}\left((\mathbf{Q} \odot \mathbf{B} \odot \mathbf{W}), \dim = 1\right)\big)\mathbf{V}, \tag{A38}$$

where $\mathbf{W}_{t,:} = \boldsymbol{w}_{\mathrm{aug}}$ and $\mathrm{sum}(\mathbf{Q}, \dim = 1)$ means calculate sum for each row of matrix $\mathbf{Q}$. Other marks are defined same as Eq. (A29).

**MetaLA block.** Regard each layer as a function mapping the input $\mathbf{X} \in \mathcal{R}^{n \times d}$ to the output $\mathbf{Y} \in \mathcal{R}^{n \times d}$ where:

$$\mathbf{X} = \begin{bmatrix} \mathbf{x}_1 \\ \mathbf{x}_2 \\ \vdots \\ \mathbf{x}_n \end{bmatrix}, \mathbf{Y} = \begin{bmatrix} \mathbf{y}_1 \\ \mathbf{y}_2 \\ \vdots \\ \mathbf{y}_n \end{bmatrix}, \tag{A39}$$

following Transformer structure, now the Token Mix mechanism Eq. (A31)-Eq. (A37) can be integrated as follows:

$$\mathbf{Y} = \text{TokenMix}(\mathbf{X}), \quad (A40)$$

and the Channel Mix part (GLU [3]) can be written as follows:

$$\mathbf{Y} = (\text{Swish}(\mathbf{X}\boldsymbol{W}_1) \odot \mathbf{X}\boldsymbol{W}_2)\boldsymbol{W}_3, \quad (A41)$$

which can be integrated as:

$$\mathbf{X}^{l+\frac{1}{2}} = \text{TokenMix}^l(\text{XNorm}^l(\mathbf{X}^l)) + \mathbf{X}^l, \quad (A42)$$

$$\mathbf{X}^{l+1} = \text{ChannelMix}^l(\text{XNorm}^{l+\frac{1}{2}}(\mathbf{X}^{l+\frac{1}{2}})), \quad (A43)$$

where $\mathbf{X}^l$ and $\mathbf{X}^{l+1} \in \mathcal{R}^{n \times d}$ refer to the input and output of block $l$ in MetaLA. We choose XNorm = LayerNorm. Stacking several MetaLA blocks above, we can derive complete MetaLA Transformer as a Linear Foundation Model, as shown in Fig. A1.

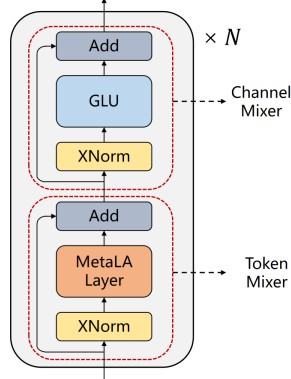

MetaLA Transformer

Figure A1: **MetaLA Transformer.** Stacking $N$ MetaLA blocks, each block is composed of two modules in sequence: token mixer and channel mixer.

## A4 Experimental Details

Table A4: **Hyper-parameters of MetaLA on LRA.** d is the dimension of model. d1 is the dimension of $d_q$ and $d_k$, d2 is the hidden dimension in GLU. num-warmup and max-step are used for cosine warmup.

| Task | Depth | d | d1 | d2 | dropout | lr | bs | wd | num-warmup | max-step |
|---|---|---|---|---|---|---|---|---|---|---|
| Listops | 6 | 32 | 16 | 64 | 0.0 | 0.0005 | 128 | 0.01 | 5000 | 50000 |
| Text | 4 | 128 | 64 | 256 | 0.1 | 0.004 | 64 | 0.0 | 10000 | 50000 |
| Retrieval | 2 | 256 | 128 | 512 | 0.1 | 0.0008 | 128 | 0.0001 | 312 | 50000 |
| Image | 6 | 512 | 256 | 512 | 0.0 | 0.003 | 128 | 0.0 | 30000 | 50000 |
| Pathfinder | 6 | 128 | 64 | 128 | 0.0 | 0.002 | 256 | 0.0 | 50000 | 500000 |
| Path-X | 6 | 32 | 16 | 32 | 0.0 | 0.00075 | 256 | 0.0 | 15000 | 500000 |

**Associative Recall (AR).** Following [36], we train two layer models with a Transformer backbone that interleaves token mixer and channel mixer. Learning rates are swept by $\text{np.logspace}(-4, -2, 4)$ for sequence length 256 and additional $\text{np.logspace}(-5, -3, 4)$ for length 512, and maximum test accuracy is reported. For GLA [15] and MetaLA, we set $H = 2$ and $d_k = d_v = d$. The kernel size of short convolution of MetaLA is 2.

**Language Modeling.** For 360M/1.4B model, we train it from scrach with a total of 15B/300B tokens on 16/32 A100 GPUs at a learning rate of 3e-4/2e-4 with batch size 0.5M/2M. Both models maintain a length of 2048 and are trained using fp16. The training setup of baselines [15, 16, 44] of 360M MetaLA are aligned with MetaLA configurations. For the 1.3B MetaLA, we compare it with publicly available models [14, 15, 16, 20, 44]. To maintain a fair comparison between linear models, we trained Mamba from scratch using the same settings with MetaLA on 100B tokens. For GLA and Retnet, we adopted the open-source checkpoints in FLA [79]. For HGRN and Pythia, we used the official open-source checkpoints. We implement all the pretrain experiments with GPT-Neox [45]. We evaluate our models on SuperGLUE benchmark [38] and Common-Sense Reasoning benchmark including LAMBADA [80], LogiQA [81], Winograd Schema Challenge (WSC273) [82], BoolQ [83], PiQA [84], HellaSwag [85], WinoGrande [86], ARC-easy (ARC-e), ARC-challenge (ARC-c) [87], OpenBookQA [88]. We report perplexity (ppl) and accuracy (acc) on LAMBADA, accuracy normalized by length on HellaSwag, ARC-challenge and OpenbookQA, and acc on the other subtasks. For SuperGLUE benchmark, we report F1 score on CB, Exact-Match (EM) score on MultiRC, and accuracy on the other subtasks, following the original work. The LM evaluation harness [89] is used to implement all evaluations.

**Long Range Arena.** We evaluate the long sequence modeling capability of our model on LRA task. We use the adamW optimizer and cosine warmup scheduler. We set the head size be 4 for group

normalization. In Retrieval, Image, Pathfinder and Path-X we use a bid-model. The hyperparameters for all tasks can be found in Table A4

**Image Classification.** We evaluate our models on ImageNet [40]. The input size of ImageNet is 224 $\times$ 224. Following Deit [49], the batch size is set to 2048 during 300 training epochs with a cosine-decay learning rate whose peak value is $2.4 \times 10^{-3}$. We choose AdamW ($\beta_1 = 0.9, \beta_2 = 0.98$) with 0.05 weight decay as the optimizer. Note that we do not use cutmix or mixup during the training.

Table A5: **Performance Comparison on Additional subtasks for Common-Sense Reasoning.** PS: parameter size (billion). T: tokens (billion). $^\dagger$ means the results reported by [20]. $^\ddagger$ indicates testing using open-source checkpoints. For baselines that need to be compared, if they do not have public checkpoints, we train and test them under identical conditions with MetaLA. LAMB: Lambada. HS: HellaSwag. WG: WinoGrande. OBQA: OpenbookQA. MetaLA$_a$: MetaLA with tied embedding trained using 100B tokens. MetaLA$_b$: MetaLA trained with 300B tokens. "AVG" refers to the average result on subtasks other than LAMBADA.

| Models | PS | T | LAMB ppl | LAMB acc | LOGIQA | WSC273 |
|---|---|---|---|---|---|---|
| Pythia$^\ddagger$ | 1.4 | 300 | 10.94 | 49.78 | 21.35 | 72.89 |
| HGRN$^\ddagger$ | 1 | 100 | 21.81 | 36.39 | 22.43 | 58.97 |
| Mamba | 1.4 | 100 | 14.02 | 44.44 | 22.73 | 68.50 |
| RetNet$^\ddagger$ | 1.3 | 100 | 22.65 | 36.79 | 22.73 | 63.74 |
| GLA$^\ddagger$ | 1.3 | 100 | 20.05 | 39.92 | 21.81 | 63.00 |
| MetaLA$_a$ | 1.3 | 100 | 12.84 | 45.99 | 21.81 | 65.93 |
| MetaLA$_b$ | 1.4 | 300 | 10.06 | 50.42 | 21.35 | 73.63 |

| Models | PS | T | BOOLQ | PIQA | HS | WG | ARC-e | ARC-c | OBQA | AVG |
|---|---|---|---|---|---|---|---|---|---|---|
| Pythia$^\ddagger$ | 1.4 | 300 | 63.12 | 70.89 | 51.98 | 56.99 | 60.56 | 28.41 | 33.20 | 51.04 |
| HGRN$^\ddagger$ | 1 | 100 | 58.75 | 71.00 | 48.05 | 51.14 | 55.51 | 28.07 | 31.80 | 47.30 |
| Mamba | 1.4 | 100 | 53.27 | 71.44 | 48.63 | 53.59 | 58.59 | 29.01 | 31.80 | 48.62 |
| RetNet$^\ddagger$ | 1.3 | 100 | 60.21 | 69.53 | 48.39 | 53.28 | 54.17 | 26.19 | 30.80 | 47.67 |
| GLA$^\ddagger$ | 1.3 | 100 | 61.04 | 70.08 | 48.00 | 51.93 | 54.88 | 28.33 | 31.40 | 47.83 |
| MetaLA$_a$ | 1.3 | 100 | 55.50 | 70.02 | 47.32 | 55.01 | 56.90 | 27.47 | 33.00 | 48.11 |
| MetaLA$_b$ | 1.4 | 300 | 56.27 | 72.25 | 53.58 | 58.17 | 61.49 | 30.03 | 34.60 | 51.26 |

Table A6: **Results on MAD tasks.** All architectures are tested according to the MAD protocol.

| Models | Compression | Fuzzy recall | In-context recall | Memorization | Noisy recall | Selective Copy | Avg |
|---|---|---|---|---|---|---|---|
| Multihead Hyena | 47.79 | 18.01 | 97.46 | 89.48 | 98.74 | 90.81 | 73.72 |
| GLA | 37.70 | 12.45 | 91.58 | 57.05 | 92.58 | 88.63 | 63.33 |
| Mamba | 43.95 | 9.60 | 87.92 | 89.45 | 90.91 | 81.79 | 67.27 |
| MetaLA | 45.55 | 15.18 | 99.87 | 85.83 | 99.73 | 97.71 | **73.98** |

## A5   Additional Experiments

**Additional subtasks for Common-Sense Reasoning.** We extend more subtasks of Commonsense Reasoning for 1B4 MetaLA. Additional experimental results are shown in Tab. A5.

**More challenging settings for MQAR.** We evaluate some models with sequence length 512 and with more retrieval key-value pairs (80, default is 64). The attention baseline benefits from global modeling capabilities, achieving optimal results. The results in Tab. A7 show that MetaLA outperforms Mamba (does not converge under the same training conditions), and there is still a significant gap compared to transformers.

Table A7: **Results on MQAR with sequence length 512 and retrieval key-value pairs 80.**

| Models | Model dimension | Acc |
|---|---|---|
| Transformer | 64 | >99.0 |
| Mamba | 64 | 0.0 |
| MetaLA | 64 | 28.5 |
| Transformer | 128 | >99.0 |
| Mamba | 128 | 0.0 |
| MetaLA | 128 | 90.4 |

**Results on the synthetic MAD task.** we evaluate several models on MAD [41], a collection of six synthetic tasks predictive of scaling laws,

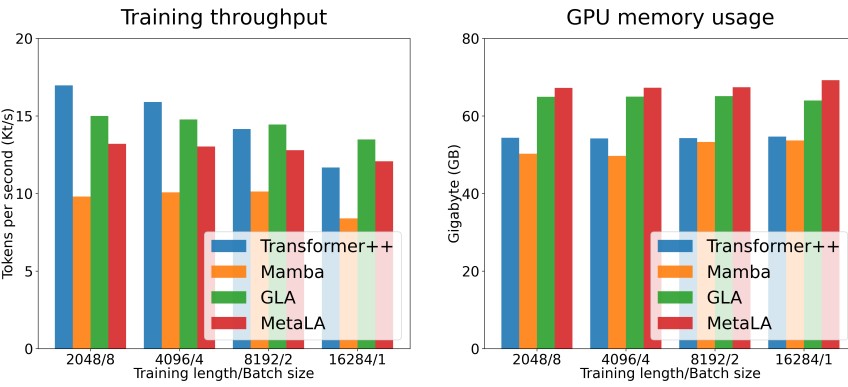

Figure A2: **Training efficiency evaluations.** The throughput and memory usage on a single A800 GPU of Transformer and various linear models. Transformer++ is implemented using FlashAttention [90] and SwiGLU.

including recall, memorization, and compression. As shown in Tab. A6, MetaLA achieves the best results across various linear complexity models.

**Results on the Needle in a Haystack (NIAH) task.** We also present experimental results on the NIAH task following [42], which is designed to evaluate the in-context retrieval capabilities of LLMs. Retrieval ability in long texts is a significant challenge for linear models, as all current linear models lack good solutions to this problem. Nonetheless, Tab. A8 shows that MetaLA has achieved satisfactory results in comparisons among linear models. Compared to Transformer models, this performance is still insufficient. This is precisely the issue we hope to address next, following the unification of linear model forms.

**The scalability with respect to model size and training tokens.** For preliminary validation, we further evaluate our model ranging in size from 380M to 3B, trained with up to 300B tokens, on the CommonSense Reasoning benchmark. The strong results in Tab. A9 demonstrate the potential of our model when scaling up the parameter scale and training dataset.

**Training efficiency evaluations.** The comparative results on training throughput and GPU memory usage across various 1.3B-sized models are shown in Fig. A2. The report indicates that: (1) Our model demonstrates good linearity, maintaining processing speed and memory efficiency with increasing sequence length, unlike the Transformer, which experiences a sharp drop in token processing speed as sequence length increases. (2) Our model matches the computational efficiency of linear models like GLA [2] in both latency and memory, and is significantly faster than Mamba [5], which also has linear complexity.

Table A8: **Results on the Needle in a Haystack (NIAH) task.** We introduce accuracy metrics across four context scales and three model scales. The middle columns display accuracies below the 4K and 8K thresholds. The rightmost columns detail both the average accuracy and the weighted average accuracy. All models are trained with sequence length 8K.

| Models | PS | Acc@2K | Acc@4K | Acc@8K | Acc@16K | Acc<=4K | Acc<=8K | Avg | Weighted Avg |
|---|---|---|---|---|---|---|---|---|---|
| LLaMA2 | 0.4 | 100.0 | 97.1 | 97.8 | 0.0 | 99.3 | 99.5 | 56.4 | 52.3 |
| HGRN2 | 0.4 | 8.6 | 6.3 | 1.3 | 0.0 | 17.0 | 9.3 | 4.9 | 4.8 |
| MetaLA | 0.4 | 25.7 | 2.9 | 11.1 | 4.4 | 11.3 | 8.7 | 8.5 | 9.0 |
| LLaMA2 | 1.0 | 100.0 | 71.4 | 73.3 | 0.0 | 92.5 | 90.9 | 47.8 | 44.1 |
| HGRN2 | 1.0 | 17.1 | 5.7 | 2.9 | 3.5 | 18.3 | 13.4 | 9.7 | 10.0 |
| MetaLA | 1.0 | 7.9 | 8.3 | 13.7 | 17.8 | 14.3 | 14.9 | 12.0 | 12.6 |
| LLaMA2 | 3.0 | 97.1 | 100.0 | 82.9 | 0.6 | 95.4 | 93.9 | 48.8 | 45.1 |
| HGRN2 | 3.0 | 58.4 | 11.4 | 2.9 | 7.3 | 46.4 | 28.9 | 18.0 | 17.9 |
| MetaLA | 3.0 | 48.3 | 7.0 | 4.1 | 18.4 | 34.8 | 22.2 | 19.1 | 19.5 |

Table A9: **Scalability tests of MetaLA on the CommonSense Reasoning benchmark.** PS: parameter size (billion). T: tokens (billion). "AVG" refers to the average result of all subtasks.

| Models | PS | T | BOOLQ | PIQA | HS | WG | ARC-E | ARC-C | OBQA | Avg |
|---|---|---|---|---|---|---|---|---|---|---|
| LLaMA2 | 0.41 | 300 | 54.04 | 67.19 | 38.75 | 52.17 | 49.24 | 23.72 | 30.00 | 45.02 |
| Cosformer2 | 0.38 | 300 | 57.40 | 66.27 | 36.65 | 50.59 | 51.81 | 23.72 | 29.00 | 45.06 |
| MetaLA | 0.38 | 300 | 60.09 | 67.79 | 38.51 | 50.99 | 52.19 | 25.60 | 30.00 | **46.45** |
| LLaMA2 | 1 | 300 | 56.42 | 69.97 | 47.04 | 52.72 | 57.07 | 28.16 | 32.60 | 49.14 |
| Cosformer2 | 1 | 300 | 44.28 | 70.73 | 45.55 | 50.51 | 55.22 | 27.30 | 31.00 | 46.37 |
| MetaLA | 1 | 300 | 59.05 | 69.37 | 46.43 | 54.38 | 57.41 | 26.96 | 33.00 | **49.52** |
| LLaMA2 | 3 | 300 | 61.31 | 73.18 | 57.88 | 59.59 | 63.93 | 31.40 | 34.00 | 54.47 |
| Cosformer2 | 3 | 300 | 50.92 | 74.27 | 57.38 | 57.30 | 63.22 | 31.40 | 35.20 | 52.81 |
| MetaLA | 3 | 300 | 62.84 | 74.16 | 59.25 | 58.80 | 64.52 | 33.28 | 35.80 | **55.52** |

