# OpenReview forum: "MetaLA: Unified Optimal Linear Approximation to Softmax Attention Map"
_NeurIPS.cc/2024/Conference — NeurIPS 2024 oral_

### Official Review · Reviewer_LW2x · 2024-06-28

**Soundness:** 2
**Presentation:** 1
**Contribution:** 2
**Rating:** 6
**Confidence:** 4

**Summary:**

Proposes a unifying view of recent linear attention/SSM/linear RNN methods and compares these models to Softmax Attention. Proposes MetaLA to address the shortcomings of the prior methods in approximating Softmax attention. Performs experiments on MQAR, language modeling, image classification and LRA.

**Strengths:**

- Addresses an  important and relevant topic of analyzing efficient sequence models that attempt to replace Softmax Attention
- Systematically comparing  and unifying the formulations of the various linear attentions and linear RNNs/SSMs is useful
- The tables classifying similarities and differences of the models are nice
- The theoretical results in the appendix appear to be sound
- The experiments performed suggest promising results

**Weaknesses:**

- The general presentation of the "optimal linear approximation" and theoretical analysis is dense and confusing and obscures the contributions.
  - I would recommend moving the Proposition statements to the main text (the proofs can remain in the appendix). E.g. Currently I get to line 218 and am told that the key value is being dropped based on theoretical analysis performed in Appendix A2, but I would rather already have an idea of this result before getting to this point. Similarly, the "self-augmentation" change seems to come out of nowhere.
  - Once you have moved these Proposition statements to the Section 4, I would recommend rewriting/reorganizing Section 4 framed around these proposition statements

- The MQAR analysis is weak. I would like to see more difficult versions of this task (larger vocab sizes, more retrievals, longer contexts) and an attention baseline to better understand the limitations and differences between the different methods.
- I think more downstream tasks should be included that require the recall/retrieval abilities that are known to affect all these fixed state size methods (e.g. https://arxiv.org/abs/2402.01032, https://arxiv.org/abs/2402.04248, https://arxiv.org/abs/2402.18510v3 to name a few, more generally these and other works should be cited) would make the proposed approach and value of the proposed framework more convincing. This would help the reader understand how much progress as actually been made toward approximating Softmax attention with more efficient methods. Even simple needle in the haystack, passkey retrieval, or phonebook retrieval (as in the Griffin paper) would help give a better sense of this.
- In the conclusion and discussion, some potential weaknesses and questions regarding the differences between Softmax attention and other methods are mentioned and proposed as future work (model capabilities, insufficient training, or eval issues). However, I would suggest some of these questions should be better explored in this work since that would help provide a better sense of how useful the proposed framework and method is.

**Questions:**

My main questions and issues are listed above.

- Line 287: typo, "Value" is misspelled in the text

**Limitations:**

Sufficient

---

> ### Author Rebuttal · Authors · 2024-08-06
>
> Thanks for your insightful feedback. We will outline your suggestions and questions, followed by our detailed responses. We hope our answers address your concerns.
>
> >*Weakness1:* The general presentation of the "optimal linear approximation" and theoretical analysis is dense and confusing and obscures the contributions. I would recommend moving the Proposition statements...
>
> **A:** We sincerely apologize for any confusion in the organization of this section. We carefully considered and discussed your suggestions, and thank you for your kind advice. We find ourselves in a bit of a dilemma. On one hand, we strive to maintain the completeness and rigor of the paper by including the proposition statement. On the other hand, we have to carefully consider space limitations, which may increase the difficulty for readers. We hope you can understand the balance we are striving to achieve. While we have not yet identified a perfect solution, we will make every effort to enhance the clarity of our presentation while taking your concerns into account. We greatly appreciate your understanding and patience.
>
> > *Weakness2:* The MQAR analysis is weak. The results of more difficult versions of this task and an attention baseline will be conducive to understanding.
>
> **A:**
> Based on your suggestion, we evaluate the performance of several models on more challenging settings. The attention baseline benefits from global modeling capabilities, achieving optimal results under both conditions discussed in the paper (>99.0). The additional experiments show that MetaLA outperforms Mamba (does not converge under the same training conditions), and there is still a significant gap compared to transformers:
> |Models|Seq_len|Key-Value Pairs|Model dimension| Acc|
> |-|-|-|-|-|
> |Transformer [2] |512|80|64|>99.0|
> |Mamba [4] |512|80|64|0.0|
> |MetaLA (ours)|512|80|64|28.5|
> |Transformer|512|80|128|>99.0|
> |Mamba|512|80|128|0.0|
> |MetaLA (ours)|512|80|128|90.4|
>
> For the results of longer sequences such as 1024, since it takes a long time for the linear model to converge, it is difficult for us to get the results during the rebuttal period. We sincerely ask for your understanding. As a supplement, we test another long sequence recall benchmark NIAH. Details are in the response to weakness3.
>
> >*Weakness3:* More downstream tasks that require the recall/retrieval abilities should be included.
>
> **A:**  According to your suggestions, we evaluate several models on **MAD** [1], a collection of six synthetic tasks predictive of scaling laws, including recall, memorization, and compression. As shown in the following table, MetaLA achieves the best results across various linear complexity models.
> |Models|Compression|Fuzzy recall|In-context recall|Memorization|Noisy recall|Selective Copy|Avg|
> |-|-|-|-|-|-|-|-|
> |Multihead Hyena [1]|47.79|18.01|97.46|89.48|98.74|90.81|73.72|
> |GLA [6]| 37.70|12.45|91.58|57.05|92.58|88.63|63.33|
> |Mamba [4]|43.95|9.60|87.92|89.45|90.91|81.79 |67.27|
> |MetaLA (ours)|45.55|15.18|99.87|85.83|99.73|97.71|**73.98**|
>
> Three distinct versions of the recall task and the selective copy task serve to underscore the effectiveness of our model in the domain of information retrieval. And we will cite the relevant articles.  We also present experimental results on the **Needle in a Haystack (NIAH)** task, which is designed to evaluate the in-context retrieval capabilities of LLMs:
> |Models|PS|Acc@2K|Acc@4K|Acc@8K|Acc@16K|Acc<=4K|Acc<=8K|Acc_Avg|Weighted_Acc_Avg|
> |-|-|-|-|-|-|-|-|-|-|
> |LLaMA2 [5]|0.4|100.0|97.1|97.8|0.0|99.3|99.5|56.4|52.3|
> |HGRN2 [2]|0.4|8.6|6.3|1.3|0.0|17.0|9.3|4.9|4.8|
> | MetaLA (ours)|0.4|25.7|2.9|11.1|4.4|11.3|8.7|8.5|9.0|
> |-|-|-|-|-|-|-|-|-|-|
> |LLaMA2|1.0|100.0|71.4|73.3|0.0|92.5|90.9|47.8|44.1|
> |HGRN2|1.0|17.1|5.7|2.9|3.5|18.3|13.4|9.7|10.0|
> |MetaLA (ours)|1.0|7.9|8.3|13.7|17.8|14.3|14.9|12.0|12.6|
> |-|-|-|-|-|-|-|-|-|-|
> |LLaMA2|3.0|97.1|100.0|82.9|0.6|95.4|93.9|48.8|45.1|
> |HGRN2|3.0|58.4|11.4|2.9|7.3|46.4|28.9|18.0|17.9|
> |MetaLA (ours)|3.0|48.3|7.0|4.1|18.4|34.8|22.2|19.1|19.5|
>
> Retrieval ability in long texts is a significant challenge for linear models, as all current linear models lack good solutions to this problem. Nonetheless, MetaLA has achieved satisfactory results in comparisons among linear models. Compared to Transformer models, this performance is still insufficient. This is precisely the issue we will address next, following the unification of linear model forms.
>
> >*Weakness4:* Some of these questions should be better explored in this work .
>
> **A:** Thank you for your valuable suggestions. During the rebuttal period, we conducted validation for MQAR, MAD, and long-text tasks based on your feedback. Additionally, we supplemented pre-training from 370M to 3B model scales on 300B tokens. We conducted more experiments around the existing work in this paper, verifying the advantages and shortcomings of MetaLA. Regarding other directions mentioned in the discussion, such as the issue of sufficient training [3] and benchmark evaluation, these are based on different fields of foundational models. We fully acknowledge the research value of these directions, which are essential for improving linear foundational models and will be part of our future work.
>
> ---
> [1] Mechanistic Design and Scaling of Hybrid Architectures, In  Arxiv 2024.
>
> [2] HGRN2: Gated Linear RNNs with State Expansion, In COLM 2024.
>
> [3] Never Train from Scratch: FAIR COMPARISON OF LONGSEQUENCE MODELS REQUIRES DATA-DRIVEN PRIORS In Arxiv 2024
>
> [4] Mamba: Linear-time Sequence Modeling with Selective State Spaces, In COLM 2024.
>
> [5] Llama 2: Open foundation and fine-tuned chat models, In Arxiv 2023.
>
> [6] Gated Linear Attention Transformers with Hardware-Efficient Training, In ICML2024.

---

> > ### Comment · Reviewer_LW2x · 2024-08-10
> >
> > - Thank you for your response. I appreciate the additional recall intensive experiments. Please ensure the gap with global attention that these experiments expose for recall intensive tasks are discussed thoroughly. This discussion of limitations and the additional experiments will strengthen the paper.
> >
> > - Please also include a softmax attention baseline for the MAD tasks in a final version to provide better comparison
> >
> > I have raised my score.

---

> ### Author Response · Authors · 2024-08-12
>
> Thank you very much for your thoughtful feedback and for raising your score. We sincerely appreciate your valuable insights, and we will make sure to thoroughly discuss the gap with global attention in recall-intensive tasks as you suggested. Additionally, we will include a softmax attention baseline for the MAD tasks in the final version to enhance the comparison. Your comments have been instrumental in improving our work, and we are grateful for your time and consideration.

---

### Official Review · Reviewer_XAnv · 2024-06-28

**Soundness:** 3
**Presentation:** 3
**Contribution:** 3
**Rating:** 8
**Confidence:** 4

**Summary:**

The paper presents a theoretical analysis of existing linear attention methods such as LinFormer, SSM, and LinRNN. Building on this analysis, the authors propose a unified framework that combines the strengths of these methods. Utilizing this framework, authors develop a novel linear attention model called MetaLA. The key innovations of MetaLA include the elimination of the Key matrix, the introduction of dynamic decay for achieving dynamic memory and static approximation, and the integration of self-augmentation with short convolution.
These enhancements aim to improve the approximation accuracy of softmax attention and outperform existing linear attention methods.

**Strengths:**

- The paper presents a comprehensive unification of various linear attention models (LinFormer, SSM, LinRNN), offering a deeper understanding of their underlying mechanisms and differences.
- The proposed MetaLA model is theoretically grounded based on the proposed framework.
- The authors provide empirical evaluation of the MetaLA model on NLP tasks, demonstrating its effectiveness and robustness.

**Weaknesses:**

- The only result for long sequences provided by the authors is on the LRA benchmark. This is not sufficient to justify the scalability of the method for real LLMs trained on sequences longer than 1k tokens since LRA is an artificial and non-representative benchmark.
- Authors provide empirical results mainly for classification tasks. It is not clear how their model performs in the text generation scenario.
- The authors neither mention nor present any experiments with low-precision formats like FP16 and BF16. Many linear attention methods fail when trained in these popular formats.
- The authors do not provide any measurements or comparisons of computational efficiency and memory requirements of their method compared to softmax attention and other linear methods.
- Since the authors do not provide code for the MetaLA model, it is hard to assess the practicality and ease of implementation.

**Questions:**

- How does MetaLA perform on tasks with longer sequences compared to standard softmax attention and other linear attention mechanisms? For example, on NarrativeQA or other tasks from LongBench.
- Can the authors provide more information on the setting of hyperparameters for their method, specifically whether the decay factor should be selected by the user?
- Another linear attention work [1], which the authors did not cite, proposes evaluating the concentration ability of linear attention using entropy and spectral gap metrics. What is the authors' opinion on these concentration metrics in the context of the MetaLA framework analysis?


[1] Linear Log-Normal Attention with Unbiased Concentration

---

> ### Author Rebuttal · Authors · 2024-08-05
>
> Thanks for your insightful feedback and your time in reading our paper.
>
>  >*Weakness1:* More long sequence validation besides the LRA benchmark. This is not sufficient to justify the scalability of the method for real LLMs.
>
> **A:** Thank you for the suggestion. In addition to the LRA benchmark, we address longer token sequences in LLM pre-training. Specifically, we pre-train a MetaLA-based language model with token counts ranging from 15B to 300B and a sequence length of **2048. To demonstrate MetaLA’s scalability, we also extended the 3B scale model experiment to 300B tokens on the CommonSense Benchmark as follows**, where MetaLA showed strong performance.
>
> |Models|PS|T| BOOLQ| PIQA |HS| WG| ARC-E| ARC-C|ОBQA|Avg|
> |-|-|-|-|-|-|-|-|-|-|-|
> |LLaMA2 [8]|0.41|300|54.04|67.19|38.75|52.17|49.24|23.72|30.00|45.02|
> |Cosformer2 [10]|0.38|300|57.40|66.27|36.65|50.59|51.81|23.72|29.00|45.06|
> |MetaLA (ours)|0.38|300|60.09|67.79|38.51|50.99|52.19|25.60|30.00|**46.45**|
> |-|-|-|-|-|-|-|-|-|-|-|
> |LLaMA2|1|300|56.42|69.97|47.04|52.72|57.07|28.16|32.60|49.14|
> |Cosformer2|1|300|44.28|70.73|45.55|50.51|55.22|27.30|31.00|46.37|
> |MetaLA (ours)|1|300|59.05|69.37|46.43|54.38|57.41|26.96|33.00|**49.52**|
> |-|-|-|-|-|-|-|-|-|-|-|
> |LLaMA2|3|300|61.31|73.18|57.88|59.59|63.93|31.40|34.00|54.47|
> |Cosformer2|3|300|50.92|74.27|57.38|57.30|63.22|31.40|35.20|52.81|
> |MetaLA (ours)|3|300|62.84|74.16|59.25|58.80|64.52|33.28|35.80|**55.52**|
>
> >*Weakness3:* About the low-precision training of FP16 and BF16.
>
> **A:** Our model is trained using fp16, and most linear models, such as [2,3,4], can be trained with half-precision, including bf16 and fp16. The numerical precision in our training shares the same characteristics as theirs.
>
> >*Weakness4:* Computational efficiency and memory requirements compared to softmax attention and other linear methods.
>
> **A:**  We report the throughput and memory footprint of 1.3B models **in the general response (please refer to the PDF)**. The findings indicate: (1) Our model demonstrates good linearity, maintaining processing speed and memory efficiency with increasing sequence length, unlike the Transformer, which experiences a sharp drop in token processing speed as sequence length increases. (2) Our model matches the computational efficiency of linear models like GLA [2] in both latency and memory, and is significantly faster than Mamba [5], which also has linear complexity.
>
> >*Weakness5:* Code Link.
>
> **A:** https://anonymous.4open.science/r/MetaLA-1BB3/README.md
>
> >*Weakness2 and Q1:* It is not clear how their model performs in the text generation scenario. How does MetaLA perform on tasks with longer sequences compared to standard softmax attention and other linear attention mechanisms? For example, on NarrativeQA or other tasks from LongBench.
>
> **A:** We have added some additional experiments based on your suggestions. We validate MetaLA in **NarrativeQA**, a natural language processing task that challenges LLMs to comprehend and answer questions about long-form narratives. The results show that MetaLA performs well on this long-sequence tasks.
>
> |Models|PS|NarrativeQA(F1)|
> |-|-|-|
> |LLaMA2 [8] |3.0|20.8|
> |Mamba [5]|3.0|17.9|
> |MetaLA (ours)|3.0|18.2|
>
> >*Q2:* About the hyperparameters.
>
> **A:** Our pre-training hyperparameter settings were based on the experience summarized by predecessors [6]. The majority of the settings are identical to those used in Pythia. Our hyperparameter settings are provided in the code link as a YML file, based on the parameter passing format used in gpt-neox [7].
>
> >*Q3:* Relathionship between the model in paper "Linear Log-Normal Attention with Unbiased Concentration" and the general linear attention model in this work.
>
> **A:** This paper and our work are complementary. In lines 633-645, we discuss that the approximation of softmax attention can be divided into two parts: value approximation and functional approximation. Our work primarily focuses on functional approximation, while the paper 'Linear Log-Normal Attention with Unbiased Concentration' that you mentioned belongs to value approximation.
>
> The paper 'Linear Log-Normal Attention with Unbiased Concentration' is a commendable work, offering extensive mathematical derivations and identifying the Log-Normal method as suitable for linear models to approximate the softmax attention map from the perspectives of entropy and spectral angles. This method has demonstrated strong performance in language models, particularly on the GLUE benchmark, and highlighted the linear complexity advantage of Linear Attention in long-sequence modeling. Our paper, on the other hand, unifies the token mixing mechanisms of Linear Attention, Linear RNN, and State Space Model, linking them to the softmax attention map from the viewpoint of attention maps. We analyze under what circumstances these models can functionally approximate softmax attention.
>
> Thus, both papers offer complementary insights into the theoretical analysis of the softmax attention map from different perspectives, indicating great potential for their combination. We are genuinely inspired by this work and will cite the paper in our own.
>
> ---
> [1] Scrolls: Standardized Comparison over Long Language Sequences, In EMNLP 2022.
>
> [2] Gated Linear Attention Transformers with Hardware-Efficient Training, In ICML2024.
>
> [3] RWKV: Reinventing RNNs for the Transformer Era, In EMNLP 2023.
>
> [4] TransNormerLLM: A Faster and Better Large Language Model with Improved TransNormer, In Arxiv 2023.
>
> [5] Mamba: Linear-time Sequence Modeling with Selective State Spaces, In COLM 2024.
>
> [6] Pythia: A Suite for Analyzing Large Language Models Across Training and Scaling, In ICML 2023.
>
> [7] https://github.com/EleutherAI/gpt-neox
>
> [8] Llama 2: Open foundation and fine-tuned chat models, In Arxiv 2023.
>
> [9] HGRN2: Gated Linear RNNs with State Expansion, In COLM 2024.
>
> [10]  You Only Scan Once: Efficient Multi-dimension Sequential Modeling with LightNet, In Arxiv 2024.

---

> ### Comment · Reviewer_XAnv · 2024-08-10
>
> Thank you for the rebuttal. The authors have addressed my concerns regarding long sequences and numerical formats. Additionally, they provided further experimental results, comparing their method to the LLaMa2 and MAMBA models. Overall I think it's a good paper and following the additional information provided by the authors, I have decided to increase my score.

---

> > ### Author Response · Authors · 2024-08-12
> >
> > Thank you sincerely for your thoughtful feedback and for increasing your score. We are grateful that our responses and additional experiments have addressed your concerns. Your support and encouragement mean a lot to us, and we truly appreciate the time you’ve taken to review our work.

---

### Official Review · Reviewer_2RWL · 2024-07-09

**Soundness:** 3
**Presentation:** 4
**Contribution:** 3
**Rating:** 7
**Confidence:** 2

**Summary:**

They proposed MetaLA, which solved problems of previous attention alternatives (LinRNN, SSM, LinFormer). They start to build their MetaLA by deriving general form of linear alternative of softmax attention.

1. They remove the K matrice redundant parameters and achieve better training efficiency
2. Add self-augmentation. This allows the input value to immediately affect the output depending on the query to prevent forgetting current token information.
3. Adding a convolutional layer at input X. This enhances the local interaction further, motivating Mamba and Griffin.

**Strengths:**

Strong build-up toward their method from well-curated baselines. Their method shows high performance empirically.

**Weaknesses:**

1. The experiment is not done with various scales. It is hard to know the effect of the scaling model and training dataset with MetaLA.
2. The performance of MetaLA is mainly shown with only benchmark scores, and there are no latency reports. Therefore, it is hard to know if this performance is Pareto optimal in the trade-off between latency and performance.

**Questions:**

1. Is approximating softmax attention the optimal solution for sequential modeling? This paper aims to approximate the various characteristics of the computational and model aspects very well. However, I wonder if softmax attention is not the optimal solution for sequential modeling. Is there any justification for it?
2. Can you provide the code?

**Limitations:**

It feels minor improvement on top of previous linear attention alternative studies.

---

> ### Author Rebuttal · Authors · 2024-08-05
>
> Thank you for your insightful feedback. Your questions are crucial and something we have been thinking about.
>
> >*Weakness1:* The experiment is not done with various scales. It is hard to know the effect of the scaling model and training dataset with MetaLA.
>
> **A:** Based on your suggestions, we conduct additional experiments. We further evaluate our model ranging in size from 380M to 3B, trained with 300B tokens, on the *CommonSense Reasoning* benchmark. Below are experimental results:
>
> |Models|PS|T| BOOLQ| PIQA |HS| WG| ARC-E| ARC-C|ОBQA|Avg|
> |-|-|-|-|-|-|-|-|-|-|-|
> |LLaMA2 [3]|0.41|300|54.04|67.19|38.75|52.17|49.24|23.72|30.00|45.02|
> |Cosformer2 [4]|0.38|300|57.40|66.27|36.65|50.59|51.81|23.72|29.00|45.06|
> |MetaLA (ours)|0.38|300|60.09|67.79|38.51|50.99|52.19|25.60|30.00|**46.45**|
> |-|-|-|-|-|-|-|-|-|-|-|
> |LLaMA2|1|300|56.42|69.97|47.04|52.72|57.07|28.16|32.60|49.14|
> |Cosformer2|1|300|44.28|70.73|45.55|50.51|55.22|27.30|31.00|46.37|
> |MetaLA (ours)|1|300|59.05|69.37|46.43|54.38|57.41|26.96|33.00|**49.52**|
> |-|-|-|-|-|-|-|-|-|-|-|
> |LLaMA2|3|300|61.31|73.18|57.88|59.59|63.93|31.40|34.00|54.47|
> |Cosformer2|3|300|50.92|74.27|57.38|57.30|63.22|31.40|35.20|52.81|
> |MetaLA (ours)|3|300|62.84|74.16|59.25|58.80|64.52|33.28|35.80|**55.52**|
>
> The experimental results corroborate the scalability and performance of MetaLA.
>
> >*Weakness2:* The performance of MetaLA is mainly shown with only benchmark scores, and there are no latency reports. Therefore, it is hard to know if this performance is Pareto optimal in the trade-off between latency and performance.
>
> **A:** Thanks for your suggestion, we report the latency/throughput results of 1.3 B models in the general response to all reviewers, please refer to the PDF file. The report indicates that:
> - Our model exhibits good linearity, maintaining processing speed with increasing sequence length, in contrast to the Transformer, which shows a sharp drop in token processing speed as sequence length increases.
> - Our model is computationally as efficient as linear models like Gated Linear Attention [1], while being significantly faster than the Mamba model [2], which also has linear complexity.
>
> >*Question1:* Is approximating softmax attention the optimal solution for sequential modeling? This paper aims to approximate the various characteristics of the computational and model aspects very well. However, I wonder if softmax attention is not the optimal solution for sequential modeling. Is there any justification for it?
>
> **A:** You raise a very profound and interesting question, one that we have been thinking about and we would love to discuss with you.
>
> **The AI (performance) perspective:** Although there is no strict theoretical proof of optimality for softmax attention, its excellent performance has been empirically validated by many large models, making it undoubtedly the foundation of existing large models.
>
> **The neuroscience (bio-plausibility) perspective:** It is almost certain that softmax attention is **NOT** the optimal solution for sequence modeling. To achieve AGI (Artificial General Intelligence), there are two constraints we cannot bypass.
>
> - **Power and Latency:** In terms of power and latency, the human brain operates very stable with bounded power consumption and trascient reaction. In contrast to Transformer models, if we assume the sequence input and output are interactions over time with the environment, then it faces a severe issue. Over time, its power consumption and latency will grow, a limitation brought by quadratic complexity.
>
> - **Physical Space Limitation:** As we pointed out in the paper, softmax attention is an infinite-state mechanism, leading to an increasing demand for physical space (for storage) as the model runs. This contradicts the human brain's ability to maintain a large and efficient memory capacity despite its finite size.
>
> It can be seen that the perspective from which we view softmax attention determines whether it is optimal, and there are trade-offs involved. Our approximation strives to restore its performance optimality (which has been empirically validated), but for aspects where softmax is not optimal, we still possess unique advantages.
>
> >*Question2:* Can you provide the code?
>
> **A:** We provide the code link here: https://anonymous.4open.science/r/MetaLA-1BB3/README.md
>
> ---
> [1] Gated Linear Attention Transformers with Hardware-Efficient Training, In ICML2024.
>
> [2] Mamba: Linear-time Sequence Modeling with Selective State Spaces, In Arxiv 2023.
>
> [3]  Llama 2: Open Foundation and Fine-tuned Chat Models, In Arxiv 2023.
>
> [4]  You Only Scan Once: Efficient Multi-dimension Sequential Modeling with LightNet, In Arxiv 2024.

---

> > ### Comment · Reviewer_2RWL · 2024-08-11
> >
> > Thank for for your thoughtfully written rebuttal. I gave the borderline accept because I could not fully understand this paper, therefore my reviewing confidence was pretty low.
> >
> > My concerns was all resolved (performance scalability and latency optimality) by general response so I want to raise my score to accept (however I am not still sure about the content of paper. E.g. equations and background theories). However, after reading the general response, now I can understand how this paper is novel and valuable in more depth.
> >
> > The response of my question about optimality of softmax attention is interesting. In my opinion, the softmax attention solves the problem of RNN and human intelligence which is limitation of memory and computation for each token. I think we should improve the softmax attention while maintaining it's non constant time and space complexity to achieve AGI that performs better than human. In this sense, I have a question about linear attention mechanisms. What if we increase memory space of linear attention mechanism as non constant (linear or log linear)? As far as I know, all linear attentions are strictly limit their space and time complexity as constant for each tokens. But I wonder that is it really pareto optimal if we scale it compare to quadratic attention. Do you think current implementation of linear attention can handle this? (Including this paper)

---

> > > ### Author Response · Authors · 2024-08-12
> > >
> > > Thank you for your feedback, suggestions, and insightful discussion, which have had a positive impact on us. We continue to address your concerns below:
> > >
> > > This is an interesting question. The current linear models can match or even outperform softmax attention in most tasks while being more efficient. However, they fall short in terms of memory capacity compared to softmax attention. As we can see, there is a conflict between the increasing input sequences and the fixed memory space. Therefore, both approaches have their advantages and disadvantages, making it difficult to achieve Pareto optimality. To develop a model that balances efficiency with good memory capability, we think that relaxing the complexity of processing each token (to log-linear) could be a promising solution. This will be left for our future work.

---

> > > > ### Comment · Reviewer_2RWL · 2024-08-13
> > > >
> > > > Thank you for kind and insightful response. I do not have any more concerns and questions anymore. It was really worth it to discuss with authors and feel honor to reviewing this paper. I also wonder what will be happened next to linear vs. quadratic attention wars. I hope your future research will reveal the parato front of these trade off soon! Again, thank you.

---

### Official Review · Reviewer_rbUC · 2024-07-13

**Soundness:** 4
**Presentation:** 3
**Contribution:** 4
**Rating:** 7
**Confidence:** 3

**Summary:**

This paper discusses the development and evaluation of linear complexity models for Transformers, which aim to replace the conventional softmax attention mechanism. The authors first unify existing models including LinFormer, SSM, and LinRNN into the framework of Linear Attention. Then they establish three (actually four) criteria for optimal linear attention: (0) linear complexity, (1) dynamic memory capability, (2) static approximation ability, and (3) minimal parameter usage. This paper shows that the three existing models mentioned above do not meet all these conditions and therefore perform suboptimally. In response, they introduce a new model, Meta Linear Attention (MetaLA), designed to fulfill these criteria. MetaLA is tested across various tasks including the Multi-Query Associative Recall (MQAR) task, language modeling, image classification, and the Long-Range Arena (LRA) benchmark, where it outperforms existing linear models, demonstrating its effectiveness.

**Strengths:**

1, This work unifies three widely-applied linear-complexity substituitions (LinFormer, SSM, LinRNN) of Transformer into the same framework, including a recurrent form and a parallel form, which enables higher-level studies and analysis.

2, This work proposes three criteria for evaluating a linear-complexity model (memory, approximation, parameter), which brings insights and targets for future developments of linear-complexity models.

3, Combining 1 and 2, this work evaluates whether each of the three existings models satisfies each criteria, and shows that every model does not fulfill all requirements. Based on this insight, they propose a MetaLA Transformer model that satisfies all three criteria and performs well in multiple tasks.

**Weaknesses:**

(After rebuttal: The authors carefully addressed my main concerns.)
--------------

Please correct me if there are any mistakes.

1, The main concern originates from the three criteria of "**optimality**" for linear attention (LA). In general, "optimality" indicates a situation that any possible method/result cannot be substantially better. However, it seems like these three criteria are all *necessary* conditions. As far as I noticed, this work has not stated whether these criteria are *sufficient* for optimality. As a consequence, even if MetaLA satisfies all three criteria, it is not proper to state that MetaLA is *optimal*.

2, The presentations could be improved, especially in Section 2. It is important to define or denote the dimensionality of not only each matrix/vector (which the authors have done) but also each function and operation (which they have not).

**Questions:**

The authors are encouraged to disagree or discuss on my questions/comments.

1, On Page 2 Line 67, what is the mask matrix $M$ like? From the context, I guess $M$ is a lower triangular matrix with all valid elements $=1$?

2, On Page 3 Line 78 (top), why does the LinFormer consume $O(1)$ time and memory complexity? Also, would you please elaborate on the "Chunkwise algorithm" mensioned in Line 80?

3, On Page 5 Line 147, why this softmax-generated expressive attention map requires infinite states?

4, On Page 5 Line 177, what is the condition (or consequence) of "good enough" inputs?

5, On Appendix A 2.3 (and also A 2.4), it is claimed that (Q, $\Lambda_t$) can meet C0, C1, C2 with fewest parameters. Is this concept of "fewest" restricted in a certain domain of model selection? In other words, do you have a lower-bound proof showing that any model utilizing fewer parameters than (Q, $\Lambda_t$) will fail?

**Limitations:**

Limitations are adequately discussed in the Section of Conclusion and Discussion.

---

> ### Author Rebuttal · Authors · 2024-08-05
>
> Thanks for your insightful feedback and your time in reading our paper.
> >*Weakness 1:*  Concerns about the optimality for linear attention.
>
> **A:** Your concern is valid. There is no definitive evidence to prove that MetaLA is the optimal approximation of self-attention. Please allow me to explain why we chose to use this term.
>
> - First, we repeatedly state in the paper that we are discussing the necessary conditions for optimality. In fact, these necessary conditions highlight an important issue, namely, what kind of models can approximate softmax attention.
>
> - Second, we use the term "optimal" to draw the community's attention to the issue of how to approximate softmax attention. We believe that better approximating softmax attention is one of the potential ways to address cutting-edge issues in linear models, such as retrieval and in-context learning.
>
> >*Weakness 2:* The presentation in Section 2.
>
> **A:** Thank you for your suggestion. We will carefully check and optimize these expressions.
>
> >*Q1:* Explanation of the mask matrix $M$.
>
> **A:** Your understanding is correct. For the matrix $M$, the lower triangular part (including the diagonal) is set to 1 to maintain model causality, ensuring that the output at time $t$ depends only on inputs before $t$. As for the upper triangular part (excluding the diagonal),
>
> - Linear attention: The upper triangular elements are set to 0, as no exponential operation is involved.
>
> - Softmax attention: The upper triangular elements are set to $-\infty$, ensuring that their exponentiated values become 0, thereby not affecting the normalization of values before $t$.
>
> >*Q2:* About the complexity of linear attention and the chunk-wise algorithm.
>
> **A:** As shown in Line 78, **during inference**, the time and memory complexity per token is $\mathcal{O}(1)$, meaning that computational and memory costs remain constant. From the parallel formula Eq.5 in Line74, we can express the inference in a recurrent form (operating separately on the numerator and denominator):
>
> $
> S_{t} = \sum_{i=1}^{t}\phi^{T}(k_{i})v_{i} = S_{t-1} + \phi^{T}(k_{t})v_{t},
> $
>
> $
> n_{t} = \sum_{i=1}^{t}\phi^{T}(k_{i}) = n_{t-1} + \phi(k_{t}),
> $
>
> $
> o_{t} = \frac{\phi(q_{t})S_{t}}{\phi(q_{t})n^{T}_{t}}.
> $
>
> In this iterative form, each operation only needs to maintain a fixed state size $S_{t}\in\mathcal{R}^{d \times d}, n_{t}\in\mathcal{R}^d$, resulting in a storage complexity of $\mathcal{O}(1)$ (occupying $d^{2}+d$ space, where $d$ is a constant). Furthermore, each token computation involves $d^{2}+d$ multiplications and additions, ensuring the time complexity is also $\mathcal{O}(1)$.
>
> **About the chunk-wise algorithm.** The Chunk-Wise algorithm combines the low complexity of recurrent operations with the high throughput of parallel processing during training. It achieves this by using serial processing between chunks and parallel processing within chunks, ensuring (1) full utilization of GPU computational units within a chunk and (2) reduced redundant calculations of historical information between chunks.
>
> For Linear Attention, normalization by the denominator is typically omitted. We modify the earlier formulas for further discussion (a more general form is provided in Section 4.2 of [1]):
>
> $
> S_{t} = S_{t-1} + k_{t}^{T}v_{t},
> $
>
> $
> o_{t} = q_{t}S_{t},
> $
>
> During training, if using a parallel form of computation, i.e., $O = (QK^{T}\odot M)V$, these matrices must be computed from left to right, resulting in a computational complexity of $\mathcal{O}(L^{2})$. To reduce the time complexity of the linear model during training, researchers use a chunk-wise algorithm, which consists of two parts: intra-chunk and inter-chunk. We divide a sequence of length $L$ into $\frac{L}{C}$ chunks (with padding if not divisible), each chunk of size $C$. The computation method is as follows(The first term on the right side of the equation is the inter-chunk, and the second term is the intra-chunk.):
>
> $
> O_{[i]} = Q_{[i]}S_{[i-1]} + (Q_{[i]}K_{[i]}^{T}\odot M)V_{[i]},
> $
>
> $
> S_{[i]} = S_{[i-1]} + K_{[i]}^{T}V_{[i]}.
> $
>
> Thus, each chunk will only generate a parallel-like computation, where the resulting $C \times C$ matrix is a submatrix of the attention map. So the computational cost is $\frac{L}{C} [(Cd^{2}+C^{2}d)+Cd^{2}]$. If $d$ and $C$ are constants, the time complexity of the computation becomes $\mathcal{O}(L)$.
>
> >*Q3:* Why self-attention requires infinite states.
>
> **A:** The computation for the $t$-th output of self-attention during inference is given by:
>
> $
> o_{t} = \text{softmax}(q_{t}K_{t}^{T})V_{t} = \frac{\sum_{i=1}^{t}\langle q_{t}, k_{i}\rangle v_{i}}{\sum_{j=1}^{t}\langle q_{t},k_{j}\rangle}
> $
>
> where $K_{t}^{T} = [k_{1}, \cdots, k_{t}]$ and $V_{t}^{T} = [v_{1}, \cdots, v_{t}]$.
>
> In the Transformer model, a KV-Cache is maintained as a state, updated after processing each input to avoid redundant computations. Upon receiving the $ (t+1) $ -th input, the state is updated by concatenating $ k_t+1$ and $ v_t+1$ to $ K_t$ and $ V_t$, respectively, which increases memory requirements over time.
>
> >*Q4:* "good enough" in line 177.
>
> **A:** In the theoretical analysis, we solved for the $Q,K,\Lambda$ matrices given any attention scores distribution. This implies a prerequisite assumption: the parameter functions $f_{q/k/\alpha}$ can project the input $X$ into the corresponding matrix space. This prerequisite imposes constraints on $f_{q/k/\alpha}$ (expressive enough) and $X$ (good enough).
>
> >*Q5:* The explanation of "fewest parameters"
>
> **A:** The minimum parameters refer to the smallest parameter group that achieves optimal approximation. If the model dimension $d, d_k, d_v$ are fixed, the minimum parameters equates to the smallest group. Please refer to line 185.
>
> ---
> [1] Gated Linear Attention Transformers with Hardware-Efficient Training, In ICML2024.

---

> > ### Comment · Reviewer_rbUC · 2024-08-12
> > **Thanks for the replies**
> >
> > The reviewer thanks the authors for your thorough and informative explanations. I am not an expert in this area, and therefore I am not able to advocate for this paper. However, I will increase my confidence from 2 to 3 as a reflection of what I have learned from your response.

---

> ### Author Response · Authors · 2024-08-12
>
> Thank you for your thoughtful feedback and for increasing your confidence in our paper. We genuinely appreciate your time and effort in reviewing our paper and responses. Your support and understanding are invaluable to us. If you have any further questions or need additional clarification, please let us know. Thank you once again for your help.

---

### Author Rebuttal · Authors · 2024-08-06

Dear ACs and Reviewers,

We would like to extend our sincere gratitude to all the reviewers for taking the time to read our paper and offering insightful suggestions. Linear models have emerged as a promising alternative to transformers, garnering significant interest within the foundational model research community. However, much of the existing work on linear models has primarily focused on application-level adaptations, such as customizing Mamba for various visual task scenarios. In contrast, our work seeks to advance the theoretical understanding of these models by contributing the following:

- **Unified linear attention.** We unify the LinFormer/SSM/LinRNN models with different origins into the form of linear attention.

- **Understanding linear attention.** We interpret the linear models as an approximation of softmax attention, and identify three necessary conditions for the optimal linear approximation.

- **Model design.** Based on the above theoretical understanding, we made three improvements to linear attention: removes the Key matrices, employs self-augmentation and exploits short convolutions.

We hope that our contributions to the unified linear attention framework and our insights into optimal approximation will inspire further principled exploration in this domain and foster the development of more effective linear models.

In response to the reviewers' suggestions, we have also conducted additional experiments:

- **Scalability.** We have extended the model to a 3B parameter scale and a 300B data scale for preliminary validation and validated it on Commensense Reasoning benchmarks. The results confirm MetaLA's scalability and efficacy.

- **Retrieval and long context abilities.** We evaluated MetaLA's retrieval performance on the MAD and MQAR tasks, and its effectiveness in handling long contexts on NarrativeQA and the Needle in a Haystack (NIAH) task.

- **Training efficiency.**  In the accompanying PDF, we provide comparative figures on training throughput and GPU memory usage across various 1.3B-sized models.

We have addressed each reviewer’s comments in detail and hope that our responses clarify any concerns. Should there be any further questions, please do not hesitate to contact us.

---

### Decision · Program_Chairs · 2024-09-25

**Decision:**

Accept (oral)

**Comment:**

This submission receives scores of 7, 7, 8, 6, indicating an acceptance of this submission. The reviewers generally agree that the paper presents a valuable contribution to the field of efficient attention mechanisms. The unified framework, theoretical analysis, and the proposed MetaLA model are novel and promising. The additional experiments and clarifications provided in the author's rebuttal further strengthen the paper's contributions.

It is important to address the reviewers' concerns regarding the clarity of the presentation and the scope of the experiments in the final version. A more thorough discussion of the limitations and trade-offs between MetaLA and softmax attention, particularly in recall-intensive tasks, would also enhance the paper.

Strengths:
- Unified Framework: The paper provides a valuable unified framework for understanding different linear attention models, including LinFormer, SSM, and LinRNN.
- Theoretical Analysis: The theoretical analysis of the optimal linear approximation to softmax attention is well-received.
- Novel Model: The proposed MetaLA model is seen as a novel and promising approach to improving the efficiency of attention mechanisms.
- Empirical Results: The empirical evaluation on various tasks demonstrates the effectiveness of MetaLA compared to existing linear models.

Area to Improve:
- Presentation: The presentation of the theoretical analysis and the "optimal linear approximation" concept could be improved for better clarity.
- Experimental Scope: Some reviewers suggest expanding the experiments to include more challenging tasks, longer sequences, and a wider range of model scales.
- Justification: Further justification is needed for certain design choices, such as the removal of the Key matrix and the use of self-augmentation.
- Comparison with Softmax Attention: A more in-depth discussion of the limitations and trade-offs between MetaLA and softmax attention, particularly in recall-intensive tasks, is recommended.